# Robust interface and reduced operation pressure enabled by co-rolling dry-process for stable all-solid-state batteries

Dong Ju Lee [1], Yuju Jeon [1], Jung-Pil Lee[2], Lanshuang Zhang [3], Ki Hwan Koh[1], Feng Li [1], Anthony U. Mu [1], Junlin Wu[3], Yu-Ting Chen[3], Seamus McNulty[1], Wei Tang[1], Marta Vicencio[1], Dapeng Xu[1], Jiyoung Kim[2] & Zheng Chen [1,3,4] ✉

The dry-process is a sustainable and promising fabrication method for all-solid-state batteries by eliminating solvents. However, a pragmatic fabrication design for thin and robust solid-state electrolyte (SSE) layers has not been established. Herein, we report a dry-process approach that enhances mechanical stability of SSE layers from film fabrication to cell operation. By co-rolling thick SSE and positive electrode feeds, a uniform, thin SSE layer (50 μm) and a high loading positive electrode layer (5 mAh cm$^{-2}$) with high active material ratio (80 wt%) are simultaneously achieved. This SSE-positive electrode integrated film exhibits enhanced physical properties and cyclability (> 80% retention after 500 cycles) at low stack pressure (2 MPa) compared to the freestanding counterparts, attributed to reinforced and intimate SSE-positive electrode interface constructed during co-rolling process. Additionally, an all-solid-state pouch cell with high stack-level specific energy (310 Wh kg$^{-1}$) and energy density (805 Wh L$^{-1}$) operating at 30 °C and 5 MPa is demonstrated.

All-solid-state batteries (ASSBs) are promising energy storage devices with higher energy density and improved safety compared to traditional lithium-ion batteries (LIBs)[1,2]. Intensive research efforts are focused on the development of solid-state electrolytes (SSEs)[3–6], positive electrodes[7–9], and negative electrodes[10–15], as well as mechanistic understandings[16–19]. Despite these progresses, there remains a lack of fabrication methods[20,21] that meet the criteria of sustainability, scalability, and high performance—key considerations for future manufacturing processes[22]. Currently, the wet slurry process, used in the production of LIBs, is considered a viable transition strategy for scaling up the manufacturing of ASSBs due to pre-existing equipment facilities[23,24]. However, this method involves the use of toxic solvents which not only pose safety risks[25,26], but also require solvent drying and recovery steps that are energy-intense and environmentally unsustainable[27]. Additionally, the chemical incompatibility of these organic solvents with many promising SSEs could compromise material integrity, thereby diminishing battery performance[28].

The dry-process is a promising option to eliminate solvents in the fabrication process[29,30] and is extensively investigated even in the industry for commercial battery manufacturing[31–34]. Though prior studies have demonstrated dry fabrication of ASSB components[35–37], dry-processing SSE layers remain multiple issues. Firstly, a thin SSE film is necessary to maximize the cell energy density, but poor mechanical properties of thin SSE layer increase the risk of mechanical failure during film fabrication and cell assembly[38]. Secondly, the use of polytetrafluoroethylene (PTFE) binder is problematic due to its poor electrochemical reduction stability[39]. However, as alternative binders typically exhibit inferior physical properties requiring higher binder

[1]Aiiso Yufeng Li Family Department of Chemical and Nano Engineering, University of California, San Diego, 9500 Gilman Drive, La Jolla, CA 92093, USA. [2]LG Energy Solution, Ltd. LG Science Park, Magokjungang 10-ro, Gangseo-gu, Seoul 07796, Republic of South Korea. [3]Program of Materials Science and Engineering, University of California, San Diego, 9500 Gilman Drive, La Jolla, CA 92093, USA. [4]Sustainable Power and Energy Center, University of California, San Diego, La Jolla, CA 92093, USA. ✉e-mail: zhc199@ucsd.edu

content[40,41], it is crucial to minimize the binder reliance in thin SSE layer fabrication. Lastly, a scalable and continuous fabrication design for a mechanically robust and thin SSE layer has not been realized. Therefore, these interconnected challenges call for a new processing strategy that ensures a thin and robust SSE layer, minimizes the risk of mechanical failure and reliance on binder, and employs a rational and scalable fabrication design.

Reducing operation or stack pressure is another critical challenge to the practical implementation of ASSBs[2]. Volume change of active materials during de-lithiation/lithiation can lead to void formation, resulting in loss of contact between SSE and active materials[42]. This mechanical degradation increases cell polarization, thus lowering the capacity utilization. While high stack pressures (> 50 MPa) are commonly used in laboratory testing to ensure particle-to-particle contacts, recent research has pivoted towards achieving stable cycling performance even at reduced stack pressures. Notably, viscoelastic SSEs have been reported to enhance SSE-positive electrode contacts and reduce operation pressure[5]. Moreover, ductile halide SSEs[43] and optimal positive electrode composite design[44] have shown to be effective under low-pressure operation. While these innovations represent breakthroughs in laboratory-scale pellet cells, transitioning these materials advancements into processable films and eventually into large-scale pouch cells is essential for the practical implementation of ASSBs. Thus, a fabrication design that can also construct a robust SSE-positive electrode interface is desired to reduce operation pressure in ASSBs.

In this work, we developed a co-rolling dry-process that significantly enhances the mechanical stability of a thin SSE layer from film fabrication to cell operation along with robust SSE-positive electrode interface. By roll-pressing a thick SSE feed with a positive electrode feed, a thin SSE (50 μm) and high-loading positive electrode (5 mAh cm$^{-2}$) layers were achieved simultaneously. Unlike the conventional approach, this co-rolling dry-process eliminates the necessity for fabricating a thin SSE layer in a freestanding form, preventing mechanical failures such as crack and tear. Moreover, the resulted robust and intimate SSE-positive electrode interface not only affords remarkable physical properties of a film, but also improves cell performance at reduced operation pressure compared to the freestanding counterparts. As a result, cycling stability over 500 cycles at 2 MPa with capacity retention over 80% under high positive electrode loading of 5 mAh cm$^{-2}$ was demonstrated. A high-specific-energy ASSB pouch cell, capable of reaching up to 310 Wh kg$^{-1}$ at 30 °C and 5 MPa, is achieved by employing co-rolled films. Our co-rolling dry-process establishes a sustainable and scalable fabrication approach and interface design strategy for the practical application of ASSBs.

## Results and Discussion
### Co-rolling strategy for thin and robust dry-processed SSE layer
Figure 1a, b illustrates a comparison of conventional dry-process and co-rolling dry process. In conventional dry-process, a thick SSE feed is progressively thinned through a series of roll-pressing (Fig. 1a). Assuming a linear relationship between the thickness and mechanical property of the film, the risk of mechanical failure increases as the thickness of the film decreases, which makes fabricating a thin but robust SSE film difficult (Fig. 1c). This low processability results in a freestanding SSE film that is prone to crack and tear (Supplementary Fig. 1a), which makes it less scalable and renders the cell assembly process including cutting, transferring, and stacking challenging[45]. Although the binder content can be increased to enhance the mechanical strength, this typically compromises ionic conductivity and electrochemical stability[36]. Thus, creating a thin but robust SSE layer is a bottleneck in terms of mechanical properties and processability for dry-processing ASSBs. In contrast, the co-rolling dry process roll-presses thick SSE and positive electrode feed layers together, thus reducing the total thickness without requiring a thin SSE layer in a

freestanding form (Fig. 1b). This approach significantly reduces the risk of mechanical failures compared to the conventional approach (Fig. 1c). Accordingly, an as-fabricated thin SSE co-rolled film displayed a uniform and crack-free surface of both SSE and positive electrode sides (Supplementary Fig. 1b). During the co-rolling process, a robust SSE-positive electrode interface is formed which facilitates the cell assembly process with mitigated risk of mechanical failure. Consequently, a practical ASSB can be realized with a thin SSE, high loading and active material ratio positive electrode, and high-capacity negative electrode, as well as a robust SSE-positive electrode interface for enhanced low stack pressure operation. (Fig. 1d).

### Fabrication of co-rolled film
Co-rolling dry-process is designed based on continuous roll-to-roll manufacturing, which is essential for scalable fabrication of battery components (Fig. 2a). This process involves three fabrication steps (S1-3). First, cathode active material (CAM), SSE, vapor-grown carbon fiber (VGCF), and binder are mixed to form the positive electrode feed layer. Second, SSE and binder are mixed to form the SSE feed layer which is placed on the positive electrode feed layer. Third, SSE-positive electrode feed layers are reduced until they reach the desired thickness.

Three fabrication parameters (P1-3), CAM particle size, co-rolling temperature, and reduction thickness, were studied based on the designed fabrication process to achieve optimal structure and uniformity of positive electrode and SSE layers. The co-rolled films were fabricated with different parameters and analyzed after pressing (Supplementary Fig. 2). First, two different types of CAM (large polycrystalline $LiNi_{0.8}Co_{0.1}Mn_{0.1}O_2$, NCM811, PC-NCM of 5-15 μm and small single crystalline $LiNi_{0.82}Co_{0.11}Mn_{0.07}O_2$, NCM82, SC-NCM of 3-5 μm) were compared (Supplementary Fig. 3). For SSE, $Li_6PS_5Cl$ (LPSCl) with small particle size (<1 μm) was used for positive electrode layer to minimize tortuosity, and relatively small particle size (2-5 μm) was used for SSE layer which was prepared by ball-milling (Supplementary Fig. 4). The active material ratio in positive electrode was fixed to 80 wt% for high energy density, and the binder ratio of 0.5 wt% was used to minimize the hinderance of $Li^+$ and $e^-$ transports in the positive electrode layer (Supplementary Fig. 5). After fabricating co-rolled films and pressing with fabrication pressure of 500 MPa, photos displayed a uniform surface on SSE and positive electrode sides for both large PC-NCM and small SC-NCM (Supplementary Figs. 2a, b) but scanning electron microscopy (SEM) images of positive electrode side showed cracked PC-NCM particles (Fig. 2b) and intact SC-NCM particles (Fig. 2c). As-fabricated films before press were compared to understand this difference. The film with PC-NCM showed a rough surface with obvious voids between large CAM and small SSE particles before press, which led to severe particle cracking after press (Supplementary Fig. 6). Meanwhile, the film with SC-NCM showed a smooth surface with dense particle packing before press, which formed intimate contacts without obvious cracking after press (Supplementary Fig. 7). The cracking of large PC-NCM was also confirmed in powder positive electrode composite at different fabrication pressures of 300 and 500 MPa (Supplementary Fig. 8). Since small SC-NCM also showed higher discharge capacities, better rate capability, and lower SSE-positive electrode resistance compared to large PC-NCM (Supplementary Fig. 9), small SC-NCM was used for further discussion.

Second, the co-rolling temperature determines the elongation of feeds. The co-rolling temperatures of 30 and 120 °C were studied by controlling the temperature of the rollers. A photo of SSE side of the film with 30 °C co-rolling displayed irregular spots (Supplementary Fig. 2c). From cross-sectional SEM images, the film with 30 °C co-rolling showed non-uniform layers of SSE and positive electrode, whereas that with 120 °C reduction showed uniform layers (Fig. 2d, e). This is due to thermo-mechanical properties of the binder, in which the modulus decreases by 67% from 30 to 120 °C (Supplementary Fig. 10). As a result, the feed layers can be more easily deformed at elevated

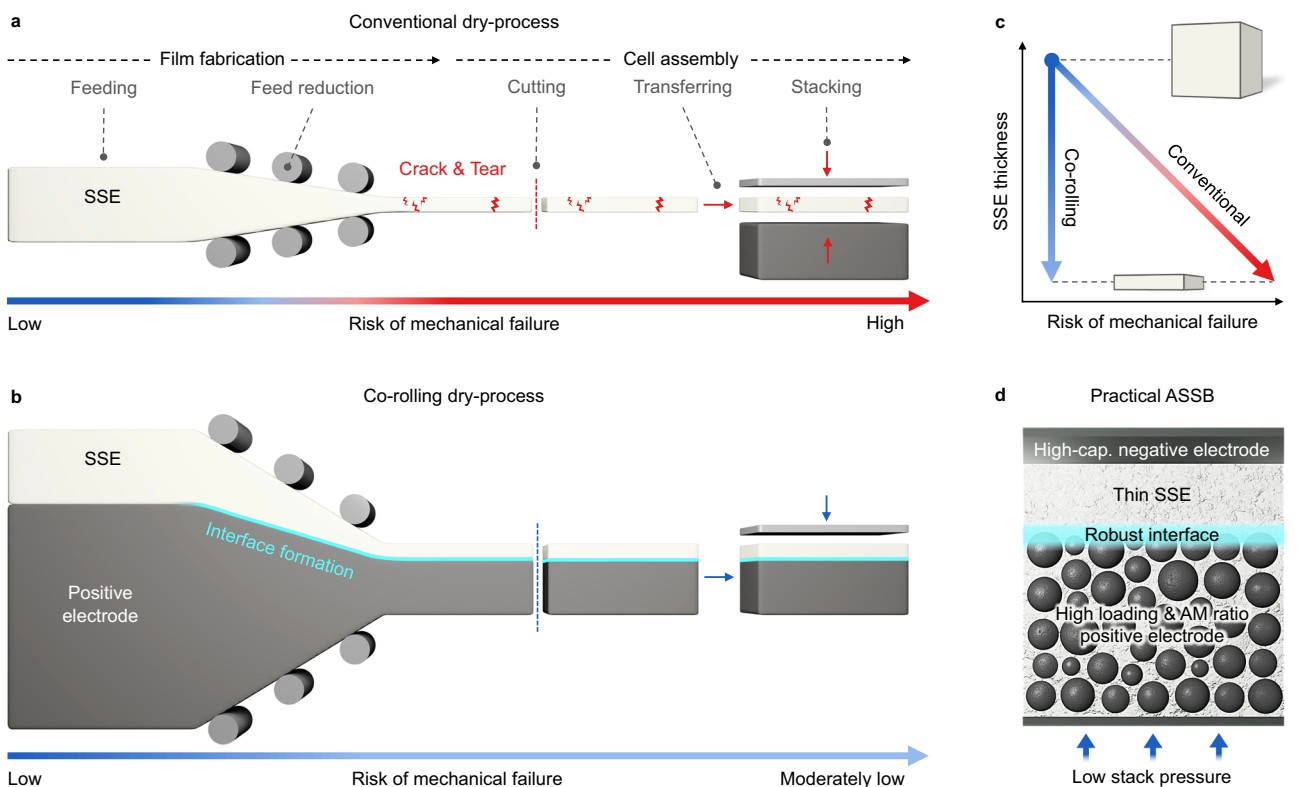

**Fig. 1 | Fabrication design of co-rolling dry-process. a, b** Schematic illustrating (**a**) conventional dry-process and (**b**) co-rolling dry-process. **c** Relation between SSE thickness and risk of mechanical failure of films. **d** Cell design and consideration for practical ASSB.

temperatures than at lower temperatures, leading to more uniform elongation of SSE and positive electrode layers during co-rolling. Third, the degree of reduction affects stress applied on feeds. The reduction thickness of 20 and 100 μm were compared by decreasing the roller gap distance by the corresponding reduction thickness every step. A photo of SSE side of the film displayed severe wrinkles for 100 μm reduction (Supplementary Fig. 2d), which was due to the penetration of positive electrode layer through SSE layer (Fig. 2f). Contrarily, the film with 20 μm reduction showed distinct layers of SSE and positive electrode without penetration (Fig. 2g), which was due to the less stress applied onto the feed layers during reduction step. Thus, co-rolling temperature and reduction thickness of 120 °C and 20 μm were used for optimal uniformity of the co-rolled film.

With these optimized fabrication parameters, a fast line speed (4 m min⁻¹) could be realized in our laboratory roller machine to fabricate a thin SSE layer compared to other published works (Supplementary Fig. 11), demonstrating its potential capability of high-throughput fabrication. Moreover, SEM images and energy dispersive X-ray spectroscopy (EDS) mapping of a co-rolled film showed a dense surface of LPSCl on SSE side (Fig. 2h), intimate coverage of SC-NCM with LPSCl on positive electrode side (Fig. 2i), and desired SSE-positive electrode interphase from a cross-section (Supplementary Fig. 12). Micro-computed tomography (CT) reconstruction of co-rolled film further confirmed the structure of SSE and positive electrode on a larger scale (Fig. 2j). The bulk and surface properties of SSE and positive electrode sides were also confirmed with X-ray diffraction (XRD) (Supplementary Fig. 13) and X-ray photoelectron spectroscopy (XPS) (Supplementary Fig. 14).

**Physical, mechanical, and electrochemical properties of co-rolled film**

The co-rolled film exhibited improved physical properties despite a thin SSE layer (50 μm). The film showed good flexibility (Fig. 3a),

recoverability (Supplementary Fig. 15) and integrity that are difficult to achieve with a thin SSE freestanding film. The tensile strength value of co-rolled film (0.510 N cm⁻¹) was approximately a sum of freestanding SSE and positive electrode films (0.049 and 0.441 N cm⁻¹) (Fig. 3b). This indicates that the mechanical property of co-rolled film is determined by both SSE and positive electrode layers instead of only SSE layer, which greatly benefits the mechanically fragile SSE layer. Compared to other published works, this co-rolling dry-process utilized the lowest binder content (<0.1 wt%) and still enabled a thin SSE layer (50 μm), demonstrating the effectiveness of this co-rolling dry-process for fabricating thin and robust SSE layer with minimal binder reliance[35,46–50] (Fig. 3c and Supplementary Table 1).

To understand the integrity of two layers, peel-off tests were conducted on freestanding and co-rolled films. Firstly, freestanding films after lamination showed completely detached SSE and positive electrode layers on the first trial of peel-off (Fig. 3d and Supplementary Movie 1). The interface structure of the freestanding films was investigated after tearing the film. A side view of freestanding films typically displayed a large gap between positive electrode and SSE layers (Supplementary Fig. 16a) and a partially attached interface that was insufficiently formed (Fig. 3e). This partially attached interface leads to limited interface adhesion for freestanding films, resulting in an easily separable interface (Fig. 3f). On the other hand, SSE layer of co-rolled film could not be detached until the tenth trial of peel-off (Fig. 3g and Supplementary Movie 2). A side view of co-rolled film also showed a well-attached SSE and positive electrode layers (Supplementary Fig. 16b). Interestingly, the positive electrode layer was still entirely attached to the SSE layer even with a slight separation that was caused by tearing the film (Fig. 3h), indicating a robust adhesion of SSE and positive electrode layers. Furthermore, a fibrillated network of binder-VGCF revealed at the separated region near the interface explains the robust adhesion of the SSE-positive electrode interface (Supplementary Fig. 17). Thus, a network-reinforced interface structure of co-rolled

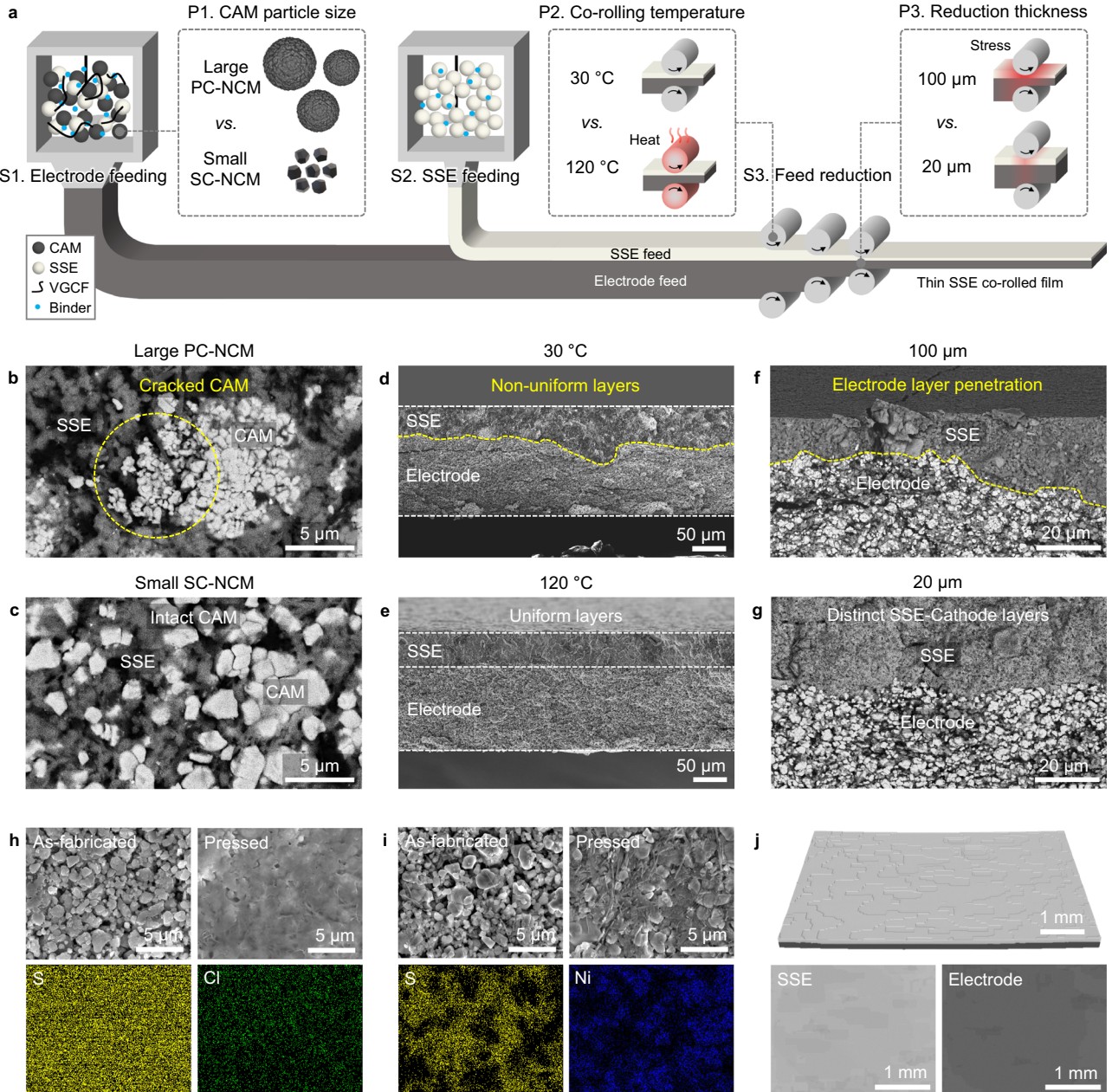

**Fig. 2 | Fabrication process and parameters of co-rolling dry-process.**
**a** Schematic illustrating three fabrication steps (S1-3) and parameters (P1-3).
**b, c** SEM image of positive electrode side of co-rolled film fabricated with (**b**) large PC-NCM and (**c**) small SC-NCM. **d, e** Cross-sectional SEM image of co-rolled film fabricated with (**d**) 30 °C and (**e**) 120 °C co-rolling temperatures. **f,** **g** Cross-sectional SEM image of co-rolled film fabricated with (**f**) 100 μm and (**g**) 20 μm reduction thicknesses. **h, i** SEM images of as-fabricated and pressed co-rolled film with optimized fabrication parameters and EDS mapping of (**h**) SSE side and (**i**) positive electrode side. **j** Micro-CT reconstruction of co-rolled film.

film enables the improved physical properties and structure integrity of SSE-positive electrode layers (Fig. 3i).

The primary difference between co-rolled film and freestanding films from a fabrication viewpoint is the degree of shear applied on the SSE-positive electrode interface. As the shear force has been reported to induce (i) fibrillation of binder[36] and (ii) contact formation of SSE particles[37] in the single-layer fabrication, such effects are extended to the interface in the dual-layer fabrication (Fig. 3j). For freestanding films, with no shear applied on the interface, (i) interface adhesion is determined by binders present on the surface of SSE and positive electrode layers, and (ii) interface contact is determined by layer-to-layer contacts made during stacking. On the other hand, for co-rolled film, with excessive shear applied on the interface during the feed reduction step, complex mechanical dynamics of the particles and binders occur at the SSE-positive electrode interface. This shear applied on the interface induces (i) movement of binder contacts yielding binder fibrillation across the interface (Supplementary Fig. 18) and (ii) formation of intimate particle-to-particle contacts at the interface yielding a fused interface for co-rolling film, after press, compared to a heterogenous interface for freestanding films (Fig. 3k, j). Therefore, co-rolling dry-process not only improves the processability of thin SSE layer, but also constructs the robust and intimate SSE-positive electrode interface structure, in both film and cell states, compared to the freestanding counterparts.

To implement this co-rolled film into a practical battery device, ensuring the desired electrochemical property of both SSE and

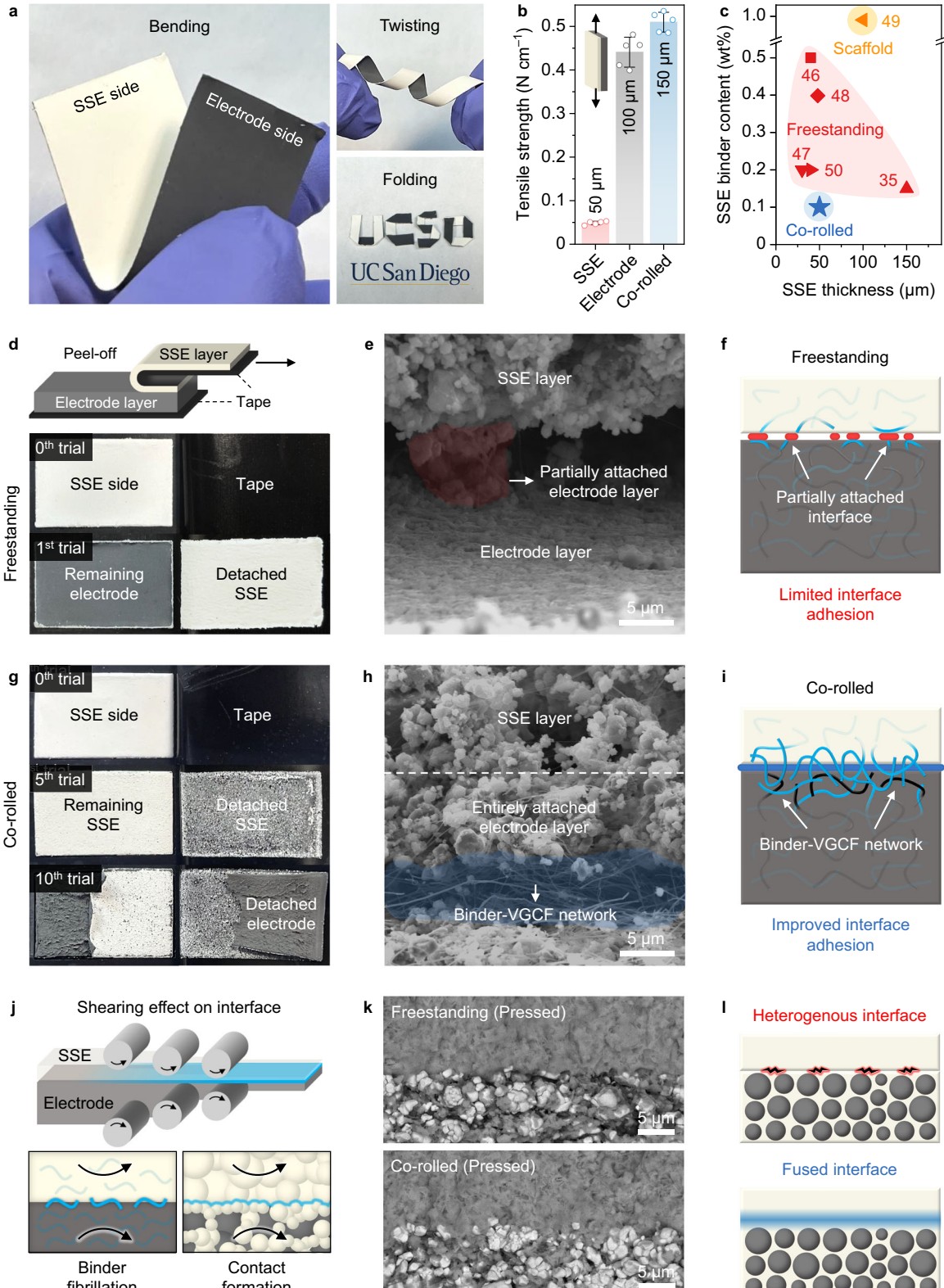

**Fig. 3 | Physical and mechanical characterizations of co-rolled film and interface. a** Flexibility test of co-rolled film. **b** Tensile strength measurement of SSE and positive electrode freestanding films and co-rolled film. Data are presented as mean values with standard deviation, $n = 5$ independent replicates. **c** Comparison of SSE binder content and SSE thickness with other published works on dry-processed SSE film[35,46–50]. **d** Peel-off test of freestanding films after lamination. **e** Side-view SEM image of freestanding films. **f** Schematic of freestanding films showing partially attached SSE-positive electrode interface and limited interface adhesion. **g** Peel-off test of co-rolled film. **h** Side-view SEM image of co-rolled film. **i** Schematic of co-rolled film showing binder-VGCF network and improved interface adhesion. **j** Schematic illustrating shearing effect on the interface: binder fibrillation and contact formation. **k** SEM images and **l** schematics of freestanding and co-rolled films showing heterogenous interface and fused interface, respectively, after press. Yellow spheres represent SSE and gray spheres represent CAM. Source data are provided as a Source Data file.

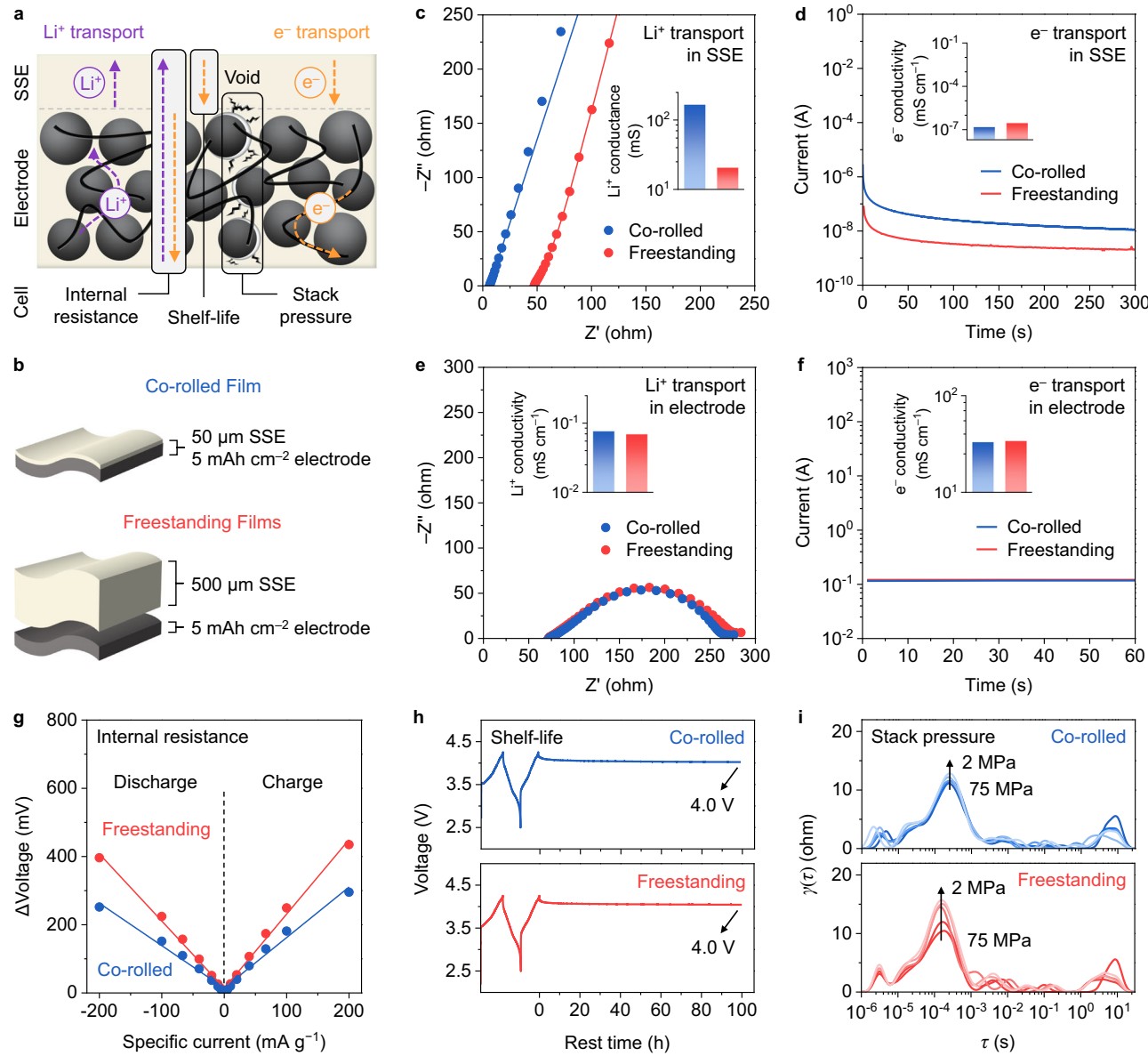

**Fig. 4 | Electrochemical characterizations of co-rolled film. a** Schematic illustrating Li⁺ and e⁻ transports in SSE and positive electrode layers, and cell properties affected by such transports. **b** Co-rolled film and freestanding films used for the characterization. **c-f** Evaluation of Li⁺ and e⁻ transports in SSE and positive electrode layers of co-rolled film and freestanding films. **c** Li⁺ transport in SSE layer. **d** e⁻ transport in SSE layer. **e** Li⁺ transport in positive electrode layer. **f** e⁻ transport in positive electrode layer. **g-i** Evaluation of cell properties of cells assembled with co-rolled film and freestanding films. **g** Internal resistance. **h** Shelf-life. **i** Stack pressure effects by DRT analysis. All tests were conducted at 23 ± 1 °C. Source data are provided as a Source Data file.

positive electrode layers is crucial. Pathways of Li⁺ and e⁻ transportation in SSE and positive electrode layers as well as their effects on a cell property are illustrated (Fig. 4a). Since their transport properties are affected by not only the composition but also structure after fabrication[51], co-rolled and freestanding films of the same compositions are fabricated and characterized with appropriate cell configurations (Fig. 4b and Supplementary Fig. 19). Note that a much thicker SSE layer of 500 μm was used for freestanding SSE film due to its limited processability, but to ensure a comparable film quality. First, Li⁺ transport in the SSE layer was characterized by electrochemical impedance spectroscopy (EIS). A much higher ionic conductance of co-rolled film than that of freestanding films (164 vs. 20 mS, corresponding to ionic conductivity of 1.04 vs. 1.29 mS cm⁻¹, respectively) was observed, which was attributed to a much thinner SSE layer (Fig. 4c, Supplementary Fig. 20a and Supplementary Table 2). Second, e⁻ transport in SSE layer was characterized by direct current

polarization (DCP). It showed comparable electronic conductivities (1.4 × 10⁻⁷ and 2.6 × 10⁻⁷ mS cm⁻¹), suggesting electron insulation (Fig. 4d). Next, Li⁺ transport in positive electrode layer presented comparable effective ionic conductivities (0.076 and 0.069 mS cm⁻¹), which implied good distribution of SSEs forming ion pathways in positive electrode layer (Fig. 4e, Supplementary Fig. 20b and Supplementary Table 3). Last, e⁻ transport in the positive electrode layer showed similar electronic conductivities (33 and 34 mS cm⁻¹), suggesting well-constructed electron pathways by VGCF in the positive electrode layer (Fig. 4f). In summary, these results confirm the desired electrochemical properties in both SSE and positive electrode layers of co-rolled film.

The electrochemical properties were further analyzed in a working cell configuration. First, the internal resistance of a cell was mainly affected by Li⁺ transport in SSE and positive electrode layers and e⁻ transport in positive electrode layer. It was characterized by applying a

pulse of different currents and measuring a change in the voltage response (Fig. 4g and Supplementary Fig. 21). The fitted slopes of polarizations[52] for co-rolled film are much lower than those of free-standing films (−1.24 vs. −1.97 for discharge and 1.48 vs. 2.18 for charge) due to the shorter ion pathway of a thinner SSE layer in co-rolled film. Second, the shelf-life of a cell is greatly affected by $e^-$ transport in the SSE layer (i.e., electron leakage) in a charged state[36,47]. The shelf-life property was evaluated by charging the cell to 4.25 V after an activation cycle and measuring cell voltage (Fig. 4h). The cell voltage for both co-rolled and freestanding films remained similar after resting for 100 h (~4.0 V), indicating a similar electron leakage through the SSE layer despite a much thinner SSE layer of co-rolled film. Last, stack pressure secures $Li^+$ and $e^-$ transport pathways against void formation during cycling. These effects were investigated with EIS and distribution of relaxation times (DRT) analysis by varying stack pressures after cycle (Fig. 4i and Supplementary Fig. 22a). Interestingly, co-rolled films showed less increase in SSE-positive electrode resistance than free-standing films with lowering stack pressure from 75 to 2 MPa. Conse-quently, while cells with both co-rolled and freestanding films showed comparable discharge capacities (~191 mAh g⁻¹) at 75 MPa (Supple-mentary Fig. 22b), a cell with co-rolled film showed a much higher discharge capacity than that with freestanding films (177 vs. 141 mAh g⁻¹) at 2 MPa (Supplementary Fig. 22c). The in situ EIS-DRT analysis further confirmed the different behavior of their resistance evolution, where the cell assembled with co-rolled film maintained lower SSE-positive electrode resistance than that with freestanding films during charge/discharge at 2 MPa (Supplementary Fig. 23). Thus, these electrochemical results imply that co-rolled film may be less susceptible to void formation and contact loss in SSE-positive elec-trode interface than freestanding films, thereby exhibiting lower resistance and delivering higher capacities at reduced stack pressure.

## Electrochemical performance of co-rolled film-based ASSBs

The electrochemical performance of co-rolled film and freestanding films were further evaluated at stack pressures of 75 and 2 MPa. It is critical to reduce the stack pressure as low as possible for practical implementation of ASSBs[2], while the low stack pressure performance is even more aggravated with increasing CAM ratio in high loading electrodes (Supplementary Fig. 24). Here, $Li_4Ti_5O_{12}$ (LTO) was used as a counter electrode (i) to isolate the pressure effects on positive elec-trode side owing to the low volume change (~0.2%) of LTO during charging and discharging[53,54] and (ii) to evaluate the performance of SSE-positive electrode structure independently by preventing the reduction of PTFE binder (<1 V vs. Li/Li⁺) within SSE layer[39]. To maintain the pressure during dynamic volume change of the positive electrode, the constant pressure setup was used for 2 MPa cycling (Supplemen-tary Fig. 25). The long-term cycling at 75 MPa showed comparable cyclability with high areal capacities of 3.55 and 3.50 mAh cm⁻² and capacity retention over 95% after 500 cycles for co-rolled and free-standing films, respectively (Fig. 5a). On the other hand, the long-term cycling at 2 MPa showed a much higher capacity for co-rolled film (3.65 vs. 3.06 mAh cm⁻²) with capacity retention over 80% after 500 cycles (Fig. 5b). All cells delivered high Coulombic efficiencies over 99.9% for 500 cycles (Supplementary Fig. 26). From the voltage profiles, the cell with co-rolled film showed a capacity loss of 8.5% at 2 MPa vs. 75 MPa (Fig. 5c), while that with freestanding films showed a higher loss of 13.2% (Fig. 5d).

To understand this difference in cycling performance at lower stack pressures, the void formation on the SSE-positive electrode interface was analyzed (Fig. 5e and Supplementary Figs. 27-29). In general, cells with co-rolled films cycled at 75 and 2 MPa showed an intimate contact at the interface between positive electrode and SSE layers (Supplementary Figs. 28a, 29a). Likewise, the cell with free-standing films cycled at 75 MPa also showed an intimate contact (Supplementary Fig. 28b); however, the cell cycled at 2 MPa showed

severe interfacial voids at the SSE-positive electrode interface (Sup-plementary Fig. 29b). The void area at different regions was further quantified (Fig. 5f). In the positive electrode layer, from 75 to 2 MPa, cells with co-rolled film showed comparable void areas (4.4 and 4.8 ratio relative to SSE layer), but those with freestanding films showed more voids (7.9 and 15.7). At the interface, from 75 to 2 MPa, cells with co-rolled film showed a slight increase (1.9 and 3.5), whereas those with freestanding films showed a significant increase (4.0 and 15.5). This result explains that the capacity loss observed with lower stack pres-sure for freestanding films can be attributed to more void formation at SSE-positive electrode interface which increases the polarization and lowers capacity utilization[55]. The less pressure change of the cell with co-rolled film during long-term cycling further supports the less for-mation of interfacial voids compared to that with freestanding films (Supplementary Fig. 30). Therefore, the intimate SSE-positive elec-trode interface of co-rolled film constructed during co-rolling process is shown to be less vulnerable to void formation at the interface, yielding an improved cyclability at reduced stack pressure compared to the freestanding counterparts.

To maximize the cell energy density, the LTO composite negative electrode should be replaced by high-capacity negative electrode. Here, the co-rolled film was coupled with 99.9 wt% Si[11] as a demon-stration of high-energy density ASSBs (Fig. 6a). The cell assembled with co-rolled film and Si stably operated up to 0.5 C without shorting at $23 \pm 1\,°C$ and 75 MPa (Fig. 6b and Supplementary Fig. 31). With both reduced SSE thickness (50 μm) and high areal loading (5 mAh cm⁻²), this cell configuration is projected to have a high specific energy of 315 Wh kg⁻¹ (Fig. 6c). In addition, a pouch cell was assembled with co-rolled film and Si (Fig. 6d) and cycled at 30 °C and stack pressure of 5 MPa over 30 cycles (Fig. 6e). The pouch cell delivered higher dis-charge capacities and specific energies than the freestanding coun-terparts (Supplementary Fig. 32) and showed a high stack-level specific energy and energy density of 310 Wh kg⁻¹ and 805 Wh L⁻¹, respectively (Supplementary Fig. 33 and Supplementary Note 1). It should be noted that further cycling induced severe current leakage due to reduction of PTFE, which requires future works on stabilizing the negative electrode interface to prevent the reduction or investigating different binders compatible with co-rolling dry-process (Supplementary Table 4). This high-energy density ASSB with co-rolled film exhibited high specific energy and energy density compared to other published works on dry-processed SSE film (Fig. 6f and Supplementary Table 1).

In this work, we conceptualized a co-rolling approach to dry-processing ASSBs that could effectively reduce the thickness of the SSE layer with minimal risk of mechanical failure compared to the con-ventional dry-process. The co-rolled film exhibited improved physical properties due to robust SSE-positive electrode interface formed during co-rolling process. This robust interface was found to be less susceptible to void formation, significantly improving cyclability (> 80% capacity retention after 500 cycles) at low stack pressure operation (2 MPa) of ASSBs compared to the freestanding counter-parts fabricated with the conventional approach. An ASSB pouch cell with high stack-level specific energy and energy density (310 Wh kg⁻¹ and 805 Wh L⁻¹, respectively) was also demonstrated by coupling with Si. This work opens a pathway for sustainable and scalable fabrication and interface design strategy for the practical application of ASSBs.

## Methods
### Fabrication of co-rolled film
For positive electrode feed layer, polycrystalline $LiNi_{0.8}Co_{0.1}Mn_{0.1}O_2$ (PC-NCM, NCM811, LG Energy Solution) or single crystalline $LiNi_{0.82}Co_{0.11}Mn_{0.07}O_2$ (SC-NCM, NCM82, MSE Supplies) (200 mAh g⁻¹), $Li_6PS_5Cl$ (LPSCl, <1 μm, vendor A (proprietary source)), and vapor-grown carbon fiber (VGCF, > 98%, Sigma-Aldrich) (80:17:3 by weight) were mixed by mortar and pestle for 30 min. The powder mixture and polytetrafluoroethylene (PTFE, <300 nm, Chemours) (100:0.5 by

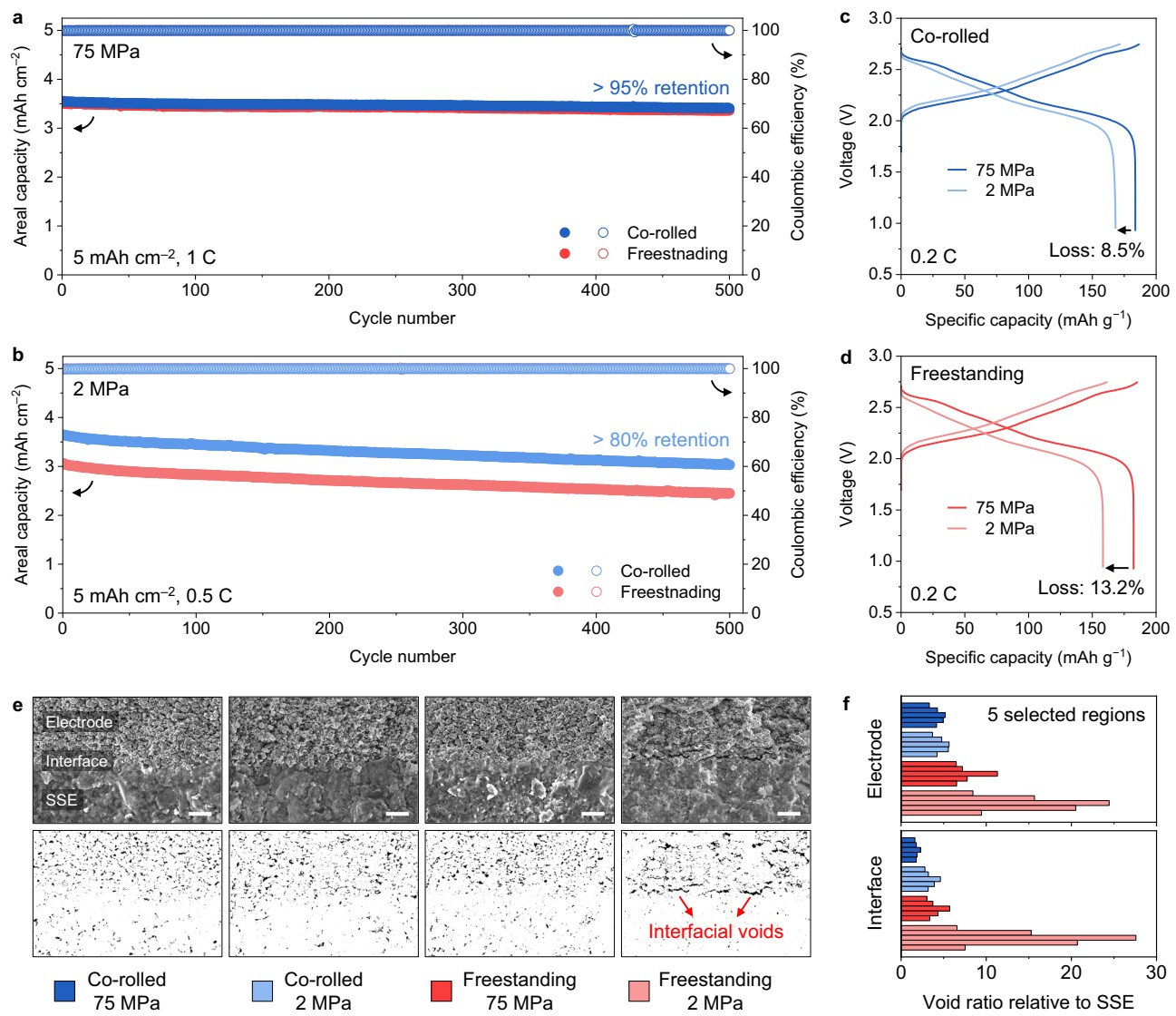

**Fig. 5 | Electrochemical performance of co-rolled film. a** Long-term cycling of co-rolled film and freestanding films at 1 C (200 mA g⁻¹) under 75 MPa and 60 °C. **b** Long-term cycling of co-rolled film and freestanding films at 0.5 C (100 mA g⁻¹) under 2 MPa and 60 °C. **c** Voltage profiles of co-rolled film at 0.2 C (40 mA g⁻¹) under 75 and 2 MPa. **d** Voltage profiles of freestanding films at 0.2 C (40 mA g⁻¹) under 75 and 2 MPa. **e** Cross-sectional SEM images and void distribution of co-rolled film and freestanding films after cycling at 75 and 2 MPa. Scale bar is 10 μm. **f** Void area of positive electrode and interface relative to SSE of co-rolled film and freestanding films after cycling at 75 and 2 MPa. For void analysis, 75 MPa cells were cycled at 1 C (200 mA g⁻¹) for 1000 cycles, and 2 MPa cells were cycled at 0.5 C (100 mA g⁻¹) for 500 cycles at 60 °C. All cells were disassembled at discharged state. Source data are provided as a Source Data file.

weight) were transferred into a 20 mL vial and vortex mixed at 3000 rpm for 3 min. The mixture was shear-mixed by mortar and pestle until a dough was formed. The dough was roll-pressed using roll-press machine (TMAXCN) at 120 °C with a fixed roller gap of 2 mm by folding and rotating, which was repeated 30 times to fibrillate PTFE binder via calendar loop[36]. For SSE layer, 2 g of LPSCl (> 95%, NEI Corporation) was ball-milled in a 50 mL zirconia jar with 75 g of zirconia balls (5 mm diameter) sealed in an Ar-filled atmosphere to reduce particle size using TMAX-PBM planetary ball miller (TMAXCN) at 400 rpm for 2 h with 1 min of intermittent rest time every 1 h. Ball-milling was conducted twice in total, after extracting and grinding, to homogenize particle size. With ball-milled LPSCl and PTFE (100:0.1 by weight), SSE feed layer was obtained using the same procedure as the positive electrode feed layer.

For co-rolled film, an areal weight ratio of positive electrode and SSE feed layers was fixed to 3.5:1 with the initial thickness of SSE feed layer to 600 μm. After cutting the feed layers into a 2.54 cm × 2.54 cm (1 in × 1 in) dimension using a hand-held punch cutter, SSE and positive

electrode feed layers are stacked and roll-pressed with desired reduction temperature (120 or 30 °C) and reduction thickness (20 or 100 μm). After reaching a feed thickness of 600 μm, the reduction thickness was reduced to 10 μm until a desired positive electrode loading was achieved. The positive electrode loading of co-rolled film was calculated and controlled based on the weight ratio of SSE feed to positive electrode feed.

## Materials characterization and image processing

Scanning electron microscopy (SEM) and energy dispersive X-ray spectroscopy (EDS) were collected using FEI Apreo SEM. SEM-EDS was carried out with a short exposure to the air at 23 ± 1 °C during sample transfer to the chamber. Cells were disassembled by gently pushing the negative electrode side of the cell with a titanium rod using hydraulic press, and cross-sections were obtained by tearing or folding the sample in Ar-filled atmosphere at 23 ± 1 °C. Micro-computed tomography (CT) was collected with Zeiss/Xradia Versa 510, and the obtained images were analyzed using Amira-Avizo software. X-ray

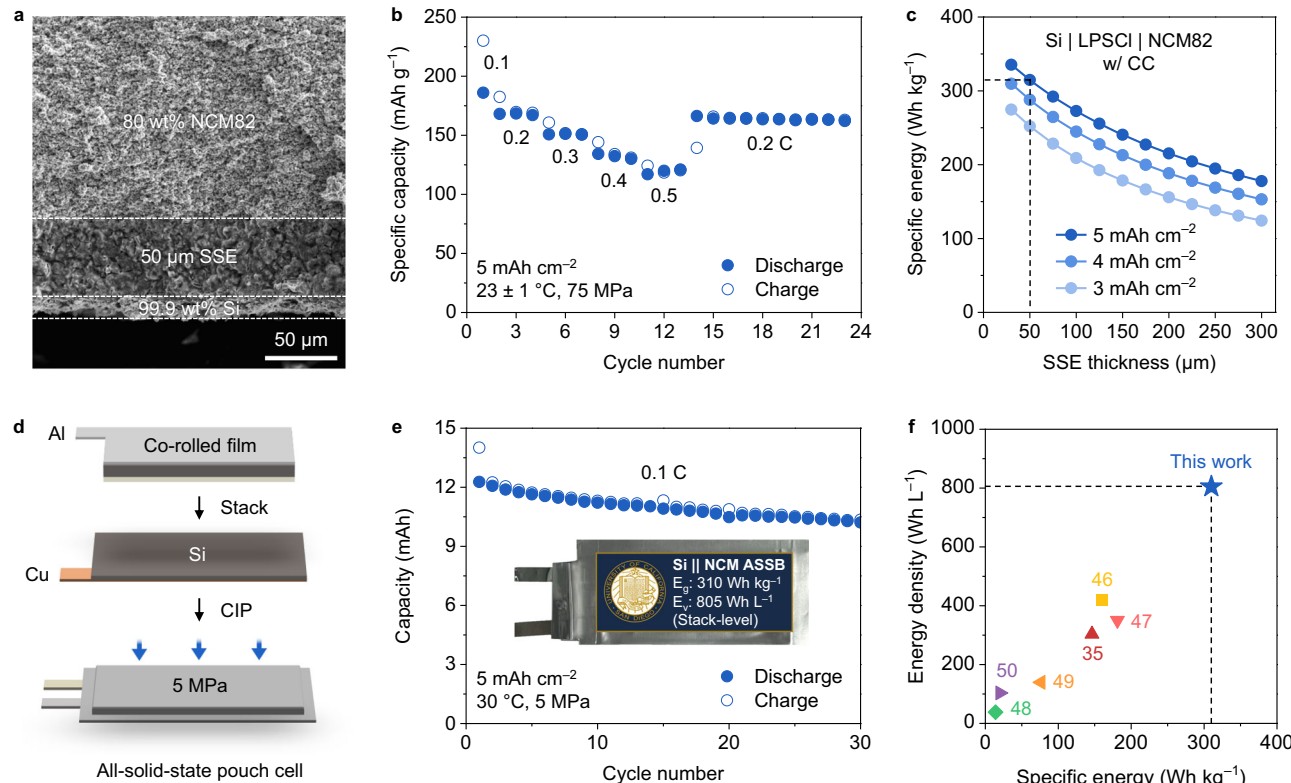

**Fig. 6 | Demonstration of high-energy density ASSBs. a** Cross-sectional SEM image of high-energy density cell assembled with co-rolled film and Si. **b** Rate test at 0.1, 0.2, 0.3, 0.4, 0.5 C (20, 40, 60, 80, 100 mA g⁻¹, respectively) under 23 ± 1 °C and 75 MPa. **c** Specific energy projection of NCM82 | LPSCl | Si cell with current collectors (CC) with varying SSE thickness and areal loading. **d** Cell assembly and fabrication of all-solid-state pouch cell with co-rolled film and Si by cold isostatic press (CIP). **e** Cycling performance of pouch cell at 0.1 C (20 mA g⁻¹) under 30 °C and 5 MPa. **f** Comparison of specific energy and energy density with published works on dry-processed SSE film[35,46–50]. Source data are provided as a Source Data file.

diffraction (XRD) patterns were collected using RIGAKU D2 Phaser with Cu Kα. X-ray photoelectron spectroscopy (XPS) was collected using AXIS Supra XPS by Kratos Analytical, and the spectra were analyzed with CasaXPS software. Dynamic mechanical analysis (DMA) was collected using Perkin Elmer DMA 8000 in the tensile mode with a temperature scan rate of 2 °C min⁻¹ and frequency of 1 Hz. Tensile strength of film was obtained using a MARK-10 M5-05 force sensor in Ar atmosphere. Void segmentation was obtained with ImageJ software[56] by adjusting the threshold of SEM images to a lower 5% of the total distribution and measuring area fraction.

### Electrochemical characterization

Electrochemical impedance spectroscopy (EIS) was performed with an applied potential of 10 mV from 7 MHz to 0.1 Hz by recording six data points per decade of frequency using Bio-Logic VSP-300. The measurements were conducted at quasi-stationary potential by applying open-circuit voltage time longer than 30 min. Direct current polarization (DCP) was performed with potential biases of 0.5 V and 0.05 V for e⁻ transports in SSE layer and positive electrode layer, respectively, using the same equipment. EIS results were fitted using ZView software. Distribution of relaxation times (DRT) was performed by using techniques previously reported[57]. Internal resistance and shelf-life were collected using Neware Battery Cycler. For Li⁺ and e⁻ transports in SSE, co-rolled film or positive electrode and SSE freestanding films were cut into a 10 mm diameter using a handheld punch cutter with SSE side facing a die plate, placed in a 10 mm polyether ether ketone (PEEK) die and pressed at 500 MPa for 3 min with titanium plungers to form SSE|positive electrode configuration. For Li⁺ transport in the positive electrode, 60 mg of LPSCl was

pressed at 300 MPa for 10 sec to make a supporting layer, and co-rolled film or freestanding positive electrode film and 60 mg of LPSCl were pressed at 500 MPa for 3 min to form SSE|positive electrode| SSE layers. 50 mg of Li₀.₅In (LiIn) powder, prepared by vortex mixing Li metal powder (FMC) and In metal powder (99.99%, MSE Supplies) of a stoichiometric ratio, was spread on both sides and pressed at 125 MPa for 1 min to form LiIn|SSE|positive electrode|SSE|LiIn electron-blocking cell configuration. For e⁻ transport in positive electrode layer, co-rolled film with SSE layer peeled-off or freestanding positive electrode film was pressed at 500 MPa for 3 min. Li⁺ or e⁻ conductivities were calculated using the following equation: $\sigma = \frac{L}{R \times A}$ (1) where $L$ is the thickness of SSE or positive electrode layer, $R$ is the resistance obtained from EIS or DCP measurements, and $A$ is the cell area. For EIS, R1 was used for Li⁺ transport in SSE, and sum of R2 and R3 was used for Li⁺ transport in positive electrode (Supplementary Fig. 20). For DCP, resistance was obtained by using the following equation: $R = \frac{V}{I}$ (2) where $V$ is the applied potential bias, and $I$ is the current measured. Internal resistance test was conducted with Si|LPSCl|NCM82 configuration by cycling at 0.1 C from 2.5 to 4.25 V, charging to 3.7 V, and applying a pulse of different C-rates for 10 s and resting for 20 min. Shelf-life test was conducted with Si| LPSCl|NCM82 configuration by cycling at 0.1 C from 2.5 to 4.25 V, charging to 4.25 V, and resting for 100 h. Stack pressure test was conducted with LiIn|LPSCl|NCM82 configuration by cycling at 0.1 C and 2 MPa from 1.875 to 3.675 V, charging to 3.4 V, and conducting EIS by varying stack pressures monitored with a load cell calibrated with Instron Loadframe. All electrochemical characterizations were performed at the environmental temperature of 23 ± 1 °C and stack pressure of 75 MPa, unless specified, in Ar-filled atmosphere.

## Electrochemical performance

Si (> 99.9%, 1-5 μm, Thermo Scientific) (3500 mAh g$^{-1}$) was mixed with polyvinylidene fluoride (PVDF, HSV900, Kynar) as binder (99.9:0.1 by weight) and N-Methyl-2-pyrrolidone (NMP, 99.5%, Sigma-Aldrich) as solvent using a planetary mixer (Thinky) to form a slurry, which was cast on rough side of Cu foil (> 99.8%, 12 μm, Gelon) using Doctor blade and dried in vacuum oven at 80 °C overnight. Li$_4$Ti$_5$O$_{12}$ (LTO, 1.5-3 μm, NEI Corporation) composite was prepared by mixing LTO, LPSCl, VGCF (60:37:3 by weight) by mortar and pestle for 30 min. To fabricate Si|LPSCl|NCM82 cells, co-rolled film or freestanding SSE and positive electrode films were cut into a 10 mm diameter using a hand-held punch cutter, placed into a 10 mm PEEK die, and pressed at 125 MPa for 1 min. Si electrode and carbon-coated Al foil (> 99.9%, 16 μm, MTI Corporation) with 10 mm diameter were placed into the negative and positive electrode sides, respectively, and the cell was pressed at 500 MPa for 3 min. The cell was cycled from 2.5 to 4.25 V at 23 ± 1 °C and a stack pressure of 75 MPa. To fabricate LTO|LPSCl|NCM82 cells, LTO composite powder was placed into a PEEK die and pressed at 300 MPa for 10 sec to form a supporting layer. Co-rolled film or freestanding SSE and positive electrode films were placed into a die, and carbon-coated Cu (> 99.9%, 9 μm, MTI Corporation) and Al foils were placed into negative and positive electrode sides, respectively. The cell was pressed at 500 MPa for 3 min and cycled from 0.95 to 2.75 V at a stack pressure of 75 MPa using fixed gap setup or 2 MPa using a constant pressure setup in 60 °C climatic chamber. To fabricate a pouch cell, co-rolled film was cut into a 2.22 cm × 1.27 cm dimension using a hand-held punch cutter and placed on Si (N/P ratio of 1.4) cast on Cu foil. The edge of co-rolled film was covered with a polyethylene terephthalate (PET) frame cut into the same dimension to prevent the edge from shorting during pressurization. Al foil was placed on positive electrode side, and the cell stack was secured with Kapton tape. Al and Ni tabs were welded on Al and Cu foils, respectively. The cell was vacuum-sealed in a laminated Al bag and pressurized at 500 MPa by cold isostatic press (MTI Corporation) for 10 min. The pouch cell was cycled from 2 to 4.25 V at 30 °C in the climatic chamber and 5 MPa in an isostatic pouch cell holder[58] by pressurizing air to control stack pressure.

## Data availability

The data supporting the findings of this study are available within the article and its Supplementary Information/Source Data files. Source data are provided with this paper.

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

## Acknowledgements

This work was supported by LG Energy Solution—U.C. San Diego Frontier Research Laboratory (FRL) via the Open Innovation program. This work was partially performed at the San Diego Nanotechnology Infrastructure (SDNI) of U.C. San Diego, a member of the National Nanotechnology Coordinated Infrastructure, which is supported by the National Science Foundation (Grant ECCS-1542148) and National Science Foundation Materials Research and Engineering Center program through the U.C. Irvine Center for Complex and Active materials (DMR-2011967).

## Author contributions

D.J.L. and Z.C. conceived the idea and wrote the manuscript. Z.C. supervised the project. D.J.L. designed and carried out the experiments. Y.J., J.-P.L., K.H.K., F.L., L.Z. assisted with data analysis. Y.J., L.Z., J.W. assisted with characterization. Y.J., J.-P.L., F.L., A.U.M. revised and commented on manuscript. Y.J., J.-P.L., K.H.K., F.L., A.U.M., Y.-T.C., S.M., W.T., M.V., D.X., J.K. participated in scientific discussions.

## Competing interests

Z.C., D.J.L., Y.J., J.-P.L. and J.K. declare that patents were filed for this work through UC San Diego's Office of Innovation and Commercialization and LG Energy Solution, Ltd. The remaining authors declare no competing interests.
