## [Peer Review File · Nature Communications]

Robust interface and reduced operation pressure enabled by co-rolling dry-process for stable all-solid-state batteries

Corresponding Author: Professor Zheng Chen

Version 0:

Reviewer comments:

Reviewer #1

(Remarks to the Author)

In this article, the author conceptualized a dry-processing co-rolling approach to reduce the thickness of SSE layer and the risk of mechanical failure. The capacity of co-rolled film-based ASSB is higher than free-standing film-based ASSB under low pressure. However, there should not be significant relation between the cycle performance and stack pressure due to the low volume expansion of cathode. Moreover, the author ignores the importance of dynamic interface stability during cycling. The present manuscript cannot provide effective guidance for further research. Currently, I do not think this work can meet the requirements of Nature Communication. Some questions are listed as follow.

1. Although the pristine SSE/cathode interface is uniform, it is much more important to maintain close contact at the interface during the cycling process. It is recommended to utilize in-situ observation methods to investigate the dynamic interface evolution during the charge/discharge process.
2. Compared to the large volume expansion of Si, the volume expansion of the cathode is quite low. Therefore, the impact of stack pressure on the anode is more pronounced than on the cathode.
3. The cycle performance of the battery utilizing dry-processed SSE layer should be summarized in Supplementary Table 1. Besides, the co-rolled SSE layer is not the thinnest compared to those reported in previous literature.
4. As shown in Figure S2, the cathode in Figure S2c exhibited the most uniform surface, however it is not the optimal experimental condition. The author should give a detailed discussion. Besides, the cathode obtained from dry process is usually not particularly uniform. Has the author investigated the impact of different conditions on this issue?
5. The cross-sectional EDS mapping of the interphase between SSE and cathode should be provided to investigate the element distribution.
6. The pouch cell utilizing free-standing SSE should be used as comparison.
7. There are some typos and format errors should be corrected such as two R3 in table 3. The author should read through the manuscript carefully and fix the errors.

Reviewer #2

(Remarks to the Author)

This article provided a novel co-rolling approach to dry-processing ASSBs that can achieve an intimate contact between composite cathode and solid electrolyte. This new strategy enabled the long cyclability (> 80% capacity retention after 500 cycles) at low stack pressure (2 MPa) due to the robust interface. Also, this work shows the probability of practical application of ASSB by achieving high-energy density ASSB coupled with Si anode (315 Wh kg⁻¹ and 840 Wh L⁻¹). However, current manuscript has to be revised because authors still need to provide clearer explanation for the several missing information. In my opinion, authors should address to several questions and comments for the publication in Nature Communications.

1. In figure 2b, SEM images showed cracked PC-NCM particles. Several articles fabricated PC-NCM based cell under 400 MPa shown in reference papers [1]-[3]. The cracking of PC-NCM is not detected at as-prepared electrode in these papers. Therefore, co-rolling process and high pressure (500 MPa) might induce the mechanical cracking. Could the authors consider decreasing the fabrication pressure? Additionally, PC-NCM is able to increase the utilization of cathode because large NCM and small SSE can increase the relative particle size (λ) according to the reference paper [4]. From the perspective of composite cathode design, PC-NCM might be more suitable than SC-NCM at typical fabrication pressure (~300 MPa). Could you provide SEM images of SC- and PC-NCM fabricated at ~300 MPa along with the corresponding

voltage profile?

Here, these are 4 reference papers.

[1] Han, Yoonjae, et al. "Single-or poly-crystalline Ni-rich layered cathode, sulfide or halide solid electrolyte: which will be the winners for all-solid-state batteries?." *Advanced Energy Materials* 11.21 (2021): 2100126.

[2] Shi, Tan, et al. "Characterization of mechanical degradation in an all-solid-state battery cathode." *Journal of Materials Chemistry A* 8.34 (2020): 17399-17404.

[3] Payandeh, Seyedhosein, et al. "The effect of single versus polycrystalline cathode particles on all-solid-state battery performance." *Advanced Materials Interfaces* 10.3 (2023): 2201806.

[4] Shi, Tan, et al. "High active material loading in all-solid-state battery electrode via particle size optimization." *Advanced Energy Materials* 10.1 (2020): 1902881.

2. Considering that the capacity of cathode is 5 mAh cm⁻², it seems to be very thick. How thick is the co-rolled cathode? Moreover, the cohesion of the cathode might be weak due to its thickness even though a robust interface is formed between the cathode and SSE. Could you provide the peel test result specifically for the cathode compared to conventional slurry casting method?

3. In my opinion, conventional electron blocking cell configuration may not be applicable for co-rolled film, as I understand the configuration is LiIn/SSE/co-rolled film (cathode/SSE)/SSE/LiIn. Therefore, it is hard to measure Li⁺ transport of SSE of solely. Also, the measured value might be effective Li⁺ conductivity in composite cathode. Could you show the detailed calculation process for determining ionic conductivity? Additionally, could you provide the ionic conductivity of each SSE independently?

4. In figure 4c, authors provided conductance of both SSE. Given that the thickness of free-standing SSE is 10 times thicker than co-rolled SSE while the area remains same, authors should compare the conductivity rather than conductance. The higher resistance of free-standing SSE could be due to the longer Li⁺ transport length (500 μm) as the resistance is proportional to thickness, as shown in the equation below. Additionally, the unit of conductance should be expressed as mS or S, not mS cm⁻².

$$R = L / (\sigma \times A)$$

R is resistance, σ is conductivity, L is thickness, and A is area.

5. Page 12, line 35, authors demonstrated a high-energy density ASSB with the co-rolled film. Please show the calculation process of gravimetric and volumetric energy density of above cell.

6. Page 15, line 25, Si electrode is composed of binder and silicon without SSE in contrast to LTO electrode. The charge transfer reaction mainly occurred at the anode surface near SSE. Also, Si electrode might suffer dynamic volume change to accommodate lithium from 5 mAh cm⁻² cathode. What is the thickness of the Si electrode before and after cycling? Additionally, is there any possibility that SSE could penetrate into the anode during 500 MPa cold isostatic press?

7. In Supplementary Fig. 14, authors proposed the interfacial fibrillation mechanism during co-rolling process. The co-rolled process can form the robust interface between cathode and SSE shown in figure 3 d, g. The improved electrochemical performance shows the importance of intimate contact. However, it is insufficient to verify the proposed mechanism. Therefore, could you provide the data that can verify your mechanism? It could intensify the meaning of the robust interface for practical application of ASSBs.

Reviewer #3

(Remarks to the Author)

Dry battery technology is an emerging field that is garnering significant attention from both academia and industry due to its potential impact on the future of the all-solid-state battery industry. However, the high operational pressure required for all-solid-state batteries remains a significant barrier to their widespread adoption. This paper proposed a co-rolling dry-process technique to reduce of battery operation pressure and improve the mechanical stability of solid-state electrolyte film, from the film fabrication through to cell operation along with cathode interfaces. The thickness of solid electrolyte layer was successfully reduced to 50 μm, and a high loading cathode layer (5 mAh cm⁻²) was fabricated without observed cracks or tears. The resulting all-solid-state batteries exhibited an enhanced cycle stability of 500 cycles under 2 MPa pressure. Overall, this research offers valuable insights in reducing battery operation pressure through modified dry processes. The manuscript is well-organized and will attract broad interests in the field of all-solid-state lithium batteries. However, after a thorough review, I have identified several issues that need to be addressed before it is suitable for publication in *Nature Communications*. These issues are outlined below:

1. While I am convinced that a low external pressure (2 MPa) is sufficient during the initial stages of battery cycling, the battery undergoes significant volume changes during continuous charging and discharging. These changes will lead to the loss of contact between solid particles due to increased internal pressures (generally >2 MPa). The authors should provide additional post-mortem analyses and discuss how the co-rolling dry-process technique helps maintain a conformal interface during long-term cycling.

2. The authors mentioned that this co-rolling dry-process utilized very low binder content (< 0.1wt%) while still achieving a thin SSE layer (page 8). However, there is no evidences or discussions of the interactions between the binders and active material particles. How about the interactions and how they change under varying pressures?

3. One key limitation of dry battery technology is the production speed. The authors should address the production speed achieved with this technique. Moreover, although the authors cite examples of the dry method from academic papers, they should also include references to its application in commercial battery manufacturing, such as patents, to demonstrate its industrial relevance.

4. If the authors claim an energy density of 315 Wh kg⁻¹ and 840 Wh L⁻¹ (Figure 6e), they should provide the actual overall weight and discharge capacity of the pouch cell. Additionally, the authors should clarify why the EIS profiles of EC/PC-NCM samples exhibit different shapes in Figure S7b. The XPS peaks are also supposed to be identified in Figure S10.

5. Binder selection is critical for dry electrodes. It would be beneficial for the authors to test and discuss suitable binders for the co-rolling dry-process technique, including the pressures applied and their compatibility with different solid-state battery systems. This information would greatly assist readers in consulting related research.

Reviewer #4

(Remarks to the Author)

The authors present an insightful approach of simultaneously rolling the cathode and SSE electrolyte films of dry processed cells. The approach has yielded significant improvement in the cathode physical and mechanical features which this reviewer has struggled with. The process excellently lends itself to roll-to-roll processing and may make a significant difference in this field.

- The device is well characterized with a good combination of imaging and advanced electrochemical diagnostics.
- After careful review, I find the manuscript to be well-written with excellent flow and grammar. The manuscript presents sufficient evidence for all claims and sufficient characterization overall.

Critically inspecting the manuscript for missing information, this reviewer cannot find major flaws for the authors to address. Therefore, this reviewer believes the manuscript can be published as-is.

Version 1:

Reviewer comments:

Reviewer #1

(Remarks to the Author)

The authors revised the manuscript carefully, I recommend acceptance of this revised manuscript.

Reviewer #2

(Remarks to the Author)

The revised manuscript provides a more detailed explanation of the formation of the fused interface between SSE and cathode, emphasizing its critical role in achieving a robust interface. In-situ EIS analysis under the low stack pressure cycling shows the low resistance from the SSE-cathode interface. However, this revised manuscript also has several ambiguous explanations. Therefore, authors need to address to questions in the manuscript.

1. Authors assumed that the cathode layer exhibits relatively high electronic conductivity (> 30 mS cm⁻¹). As shown in figure 4f, the electronic conductivity in cathode is about 30 mS cm⁻¹, which is 108 times larger than that in SSE in figure 4d. This suggests that the electronic conductivity has a negligible impact on the bulk resistance observed in EIS and DCP. However, the ionic conductivity value of cathode layer cannot be negligible (~0.1 mS cm⁻¹) because the ionic conductivity of SSE is 1.29 mS cm⁻¹ through author's response. Could the authors clarify how the ionic conductivity is distinguished between SSE and cathode layer?

2. In Supplementary Fig. 23, the resistance from SSE-cathode interface is increased during charge. Authors defined the onset of resistance increase as "interface evolution". How do authors confirm that the new interface between SSE and cathode is formed? Is there any probability that the contact loss between SSE and cathode due to the volume change of cathode particles? Alternatively, were the cells operated at sufficiently fast C-rate to rule out chemical degradation effects?

3. In Supplementary Fig. 25, authors showed the pressurization test of LTO || NCM full-cell. While this experimental setup effectively demonstrates the optimized performance of the co-rolled film, I recommend including pressurization tests for freestanding films as well. This comparison could further emphasize the mechanical and operational advantages of the co-rolled films over freestanding counterparts. Given the expected more dramatic distance changes in freestanding films, such results would strongly support the importance of achieving a robust SSE-cathode interface through the co-rolling process.

Reviewer #3

(Remarks to the Author)

The authors have revised the manuscript very carefully. All comments have been well addressed. It can be accepted by Nature Commun in the current form.

Version 2:

Reviewer comments:

Reviewer #2

(Remarks to the Author)

The revised manuscript carefully addressed all comments. I recommend acceptance of this manuscript.

We are grateful to the reviewers for their valuable comments and feedback to improve our manuscript. We believe that the comments and questions raised by the reviewers have helped us clarify our arguments and show the novelty of our work. We respond to each question raised by the reviewers in blue, and highlight the changes made to our manuscript in yellow.

Reviewer #1

In this article, the author conceptualized a dry-processing co-rolling approach to reduce the thickness of SSE layer and the risk of mechanical failure. The capacity of co-rolled film-based ASSB is higher than free-standing film-based ASSB under low pressure. However, there should not be significant relation between the cycle performance and stack pressure due to the low volume expansion of cathode. Moreover, the author ignores the importance of dynamic interface stability during cycling. The present manuscript cannot provide effective guidance for further research. Currently, I do not think this work can meet the requirements of Nature Communication. Some questions are listed as follow.

Response: We first thank the reviewer for the insightful comments and suggestions which helped us to clarify our arguments and improved the quality of the manuscript. In our responses below, we have provided clarifications regarding the relation between the cycling performance and stack pressure and dynamic interface stability and evolution. We hope our responses herein provide answers to these questions.

1. Although the pristine SSE/cathode interface is uniform, it is much more important to maintain close contact at the interface during the cycling process. It is recommended to utilize in-situ observation methods to investigate the dynamic interface evolution during the charge/discharge process.

Response: We appreciate the reviewer for the constructive comment. We acknowledge that our initial version of the manuscript might have lacked sufficient detail on the dynamic interface evolution during charge and discharge process with a suitable in-situ observation method. Please allow us to clarify these aspects.

First, we wish to highlight the intrinsic difference between the SSE-cathode interface structures of co-rolled film and freestanding films. Unlike single-layer fabrication (*i.e.*, freestanding SSE and cathode films), dual-layer fabrication (*i.e.*, co-rolled SSE-cathode film) involves the excessive degree of shear applied to the SSE-cathode interface during co-rolling process (Fig. 3j). This shearing creates intimate particle-to-particle contacts, yielding intimate SSE-cathode interface (fused interface) in co-rolled film compared to less intimate SSE-cathode interface (heterogenous interface) in freestanding films (Figure 3k, l). This close interface contact is beneficial under low stack pressure cycling (*e.g.*, 2 MPa) where voids and cracks are formed and maintained during cycling.

Second, we performed in-situ EIS-DRT analysis on co-rolled film and freestanding films at 2 MPa to investigate the dynamic interface evolution during charge and discharge process (Supplementary Fig. 23). As shown in the contour mappings, interface evolution around 10^{-4} s (indicating cathode-SSE resistance) of the time constant is observed during the charge process. The cell with freestanding films shows much higher intensity of this peak than that with co-rolled film at the end of the charge, and the peak intensity is mostly maintained during the whole discharge process. The result is also consistent with the ex-situ measurement at 2 MPa (Fig. 3i). This distinct behavior of interface evolution provides strong evidence that the intimate SSE-cathode interface in co-rolled film maintains a closer contact than freestanding films during cycling as further supported in the post-mortem analysis (Fig. 5e, f).

Third, pressurization control is also crucial for low stack pressure cycling. As the reviewer pointed out, the dynamic volume change occurs during charge/discharge. We have monitored the pressure change using LTO || NCM system at stack pressures of 75 and 2 MPa (Supplementary Fig. 25). Using fixed gap setup, the pressure decreases of 1.5 and 0.3 MPa were observed for 75 and 2 MPa, respectively. This change corresponds to a 2% decrease for 75 MPa but 15% decrease for 2 MPa. Therefore, constant pressure setup was used for 2 MPa cycling to maintain consistent pressurization during long-term cycling.

In summary, the intimate interface contact could be maintained for co-rolled film at low stack pressure (1) due to the intimate and fused SSE-cathode interface after fabrication, (2) which exhibited much lower resistance than freestanding films from in-situ EIS-DRT analysis, and (3) by employing the constant pressurization setup for long-term cycling.

We have edited the manuscript as below (Page 9-10, Main text):

“The primary difference between co-rolled film and freestanding films from a fabrication viewpoint is the degree of shear applied on the SSE-cathode interface. As the shear force has been reported to induce (i) fibrillation of binder³⁶ and (ii) contact formation of SSE particles³⁷ in the single-layer fabrication, such effects are extended to the interface in the dual-layer fabrication (Fig. 3j). For freestanding films, with no shear applied on the interface, (i) interface adhesion is determined by binders present on the surface of SSE and cathode layers, and (ii) interface contact is determined by layer-to-layer contacts made during stacking. On the other hand, for co-rolled film, with excessive shear applied on the interface during the feed reduction step, complex mechanical dynamics of the particles and binders occur at the SSE-cathode interface. This shear applied on the interface induces (i) movement of binder contacts yielding binder fibrillation across the interface (Supplementary Fig. 18) and (ii) formation of intimate particle-to-particle contacts at the interface yielding a fused interface for co-rolling film, after press, compared to a heterogenous interface for freestanding films (Fig. 3k, j). Therefore, co-rolling dry-process not only improves the processability of thin SSE layer, but also constructs the robust and intimate SSE-cathode interface structure, in both film and cell states, compared to the freestanding counterparts.”

(Page 12, Main text)

“Consequently, while cells with both co-rolled and freestanding films showed comparable discharge capacities ($\sim 191 \text{ mAh g}^{-1}$) at 75 MPa (Supplementary Fig. 22b), a cell with co-rolled film showed a much higher discharge capacity than that with freestanding films (177 vs. 141 mAh g^{-1}) at 2 MPa (Supplementary Fig. 22c). The in-situ EIS-DRT analysis further confirmed the different behavior of their interface evolution, where the cell assembled with co-rolled film maintained stable interface impedance during charge/discharge at 2 MPa but that with freestanding films showed a larger increase in impedance (Supplementary Fig. 23).”

(Page 14, Main text)

“Here, $\text{Li}_4\text{Ti}_5\text{O}_{12}$ (LTO) was used as a counter electrode (i) to isolate the pressure effects on cathode side owing to the low volume change ($\sim 0.2\%$) of LTO during charging and discharging^{53,54} and (ii) to evaluate the performance of SSE-cathode structure independently by preventing the reduction of PTFE binder ($< 1 \text{ V vs. Li/Li}^+$) within SSE layer³⁹. To maintain the pressure during dynamic volume change of the cathode, the constant pressure setup was used for 2 MPa cycling (Supplementary Fig. 25).”

Fig. 3 | Physical and mechanical characterizations of co-rolled film and interface. **a** Flexibility test of co-rolled film. **b** Tensile strength measurement of SSE and cathode freestanding films and co-rolled film. **c** Comparison of SSE binder content and SSE thickness with other published works on dry-processed SSE film^{35,46–50}. **d** Peel-off test of freestanding films after lamination. **e** Side-view SEM image of freestanding films. **f** Schematic of freestanding films showing partially

attached SSE-cathode interface and limited interface adhesion. **g** Peel-off test of co-rolled film. **h** Side-view SEM image of co-rolled film. **i** Schematic of co-rolled film showing binder-VGCF network and improved interface adhesion. **j** Schematic illustrating shearing effect on the interface: binder fibrillation and contact formation. **k** SEM images and **l** schematics of freestanding and co-rolled films showing heterogenous interface and fused interface, respectively, after press. Yellow spheres represent SSE and gray spheres represent CAM.

Supplementary Fig. 23 | In-situ observation of interface evolution of co-rolled films and freestanding films at 2 MPa in a half-cell configuration using LiIn as the counter electrode. EIS measurement and DRT analysis, respectively, of **(a, b)** discharge process and **(c, d)** charge process of co-rolled film. EIS measurement and DRT analysis, respectively, of **(e, f)** discharge process and **(g, h)** charge process of freestanding films.

Supplementary Fig. 25 | Cell design and pressurization methods used to evaluate long-term cycling stability of SSE-cathode structures. **(a)** A schematic of LTO || NCM full-cell with their volume change during charging. Photos of **(b)** fixed gap setup and **(c)** constant pressure setup. Monitored pressure changes during cell operation of **(d)** 75 MPa – fixed gap, **(e)** 2 MPa – fixed gap, and **(f)** 2 MPa – constant pressure setups. For long-term cycling stability test at different stack pressures, the fixed gap setup was used for 75 MPa, and the constant pressure setup was used for 2 MPa. Note that the constant pressure setup could not be used for 75 MPa due to the lack of suitable springs to sustain such high pressure.

2. Compared to the large volume expansion of Si, the volume expansion of the cathode is quite low. Therefore, the impact of stack pressure on the anode is more pronounced than on the cathode.

Response: The reviewer raised an insightful point. It is true that the volume expansion of Si is much larger than that of the cathode active material. However, the fundamental difference between Si anode and composite cathode is the presence of solid-state electrolyte (SSE) in the electrode. For cathode, because cathode particles are surrounded by deformed SSE particles, any volume change of cathode particle will lead to non-recoverable void formation and contact loss, especially at low stack pressure. Especially at a much higher areal loading compared with anode, the overall volume change of the cathode at the electrode level becomes significant despite a relatively small specific volume expansion. This is also the reason why forming intimate contacts is crucial during the fabrication process. On the other hand, for Si anode containing 99.9wt% Si and 0.1wt% binder, lithiated Si particles can deform and form contacts if sufficient and uniform pressures are applied^{1,2}. For example, we have recently reported that 99.9wt% Si anode could be cycled stably over hundreds of cycles at 5 MPa and room temperature³. Therefore, we believe that the impact of stack pressure is also critical on the cathode. Certainly, there are other challenges to further improve the cyclability and kinetics of Si-only anode, but we would like to be clear that Si anode was used as a demonstration of high-energy density ASSB as the focus of this work is on the fabrication of SSE and cathode layers.

1. Cao, Q. *et al.* Stacking pressure homogenizes the electrochemical lithiation reaction of silicon anode in solid-state batteries. *Energy Storage Mater.* **67**, 103246 (2024).
2. Nelson, D. L. *et al.* Fracture Dynamics in Silicon Anode Solid-State Batteries. *ACS Energy Lett.* 6085–6095 (2024) doi:10.1021/acsenergylett.4c02800.
3. Chen, Y. *et al.* Enabling Uniform and Accurate Control of Cycling Pressure for All-Solid-State Batteries. *Adv. Energy Mater.* **14**, 2304327 (2024).

Another factor affecting the low stack pressure performance of the cathode is the cathode active material ratio (Supplementary Fig. 24). It was shown that capacity utilization and rate capability aggravated with increasing cathode active material ratio. Because we used a relatively high cathode active material ratio (80wt%) to maximize the energy density, this could be a reason why we observed a pronounced impact of cell performance with lower stack pressure. Our result is also consistent with the finding as previously reported⁴.

4. Gao, X. *et al.* Solid-state lithium battery cathodes operating at low pressures. *Joule* **6**, 636–646 (2022).

We have edited the manuscript as below (Page 14, Main text):

“It is critical to reduce the stack pressure as low as possible for practical implementation of ASSBs², while the low stack pressure performance is even more aggravated with increasing CAM ratio in high loading electrodes (Supplementary Fig. 24).”

Supplementary Fig. 24 | (a-c) Voltage profiles at 0.1 C (20 mA g^{-1}) and (d-f) rate test at 0.1, 0.2, 0.3, 0.5, 1 C ($20, 40, 60, 100, 200 \text{ mA g}^{-1}$, respectively) of different cathode active material ratios of 60, 70, and 80wt%, respectively, in cathode.

3. The cycle performance of the battery utilizing dry-processed SSE layer should be summarized in Supplementary Table 1. Besides, the co-rolled SSE layer is not the thinnest compared to those reported in previous literature.

Response: We thank the reviewer's suggestion and comment. We have updated cycle performance in Supplementary Table 1 as below:

Supplementary Table 1 | Comparison of published works on dry-processed SSE layer.

		Ref 35	Ref 46	Ref 47	Ref 48	Ref 49	Ref 50	This work
SSE	Material	Li ₆ PS ₅ Cl	Li ₆ PS ₅ Cl + Li ₃ InCl ₆	Li _{5.4} PS _{4.4} Cl _{1.6}	Li ₆ PS ₅ Cl	Li ₁₀ GeP ₂ S ₁₂	Li ₆ PS ₅ Cl	Li ₆ PS ₅ Cl
	Binder	PTFE	PTFE	PTFE	PTFE	PTFE-Nylon mesh	PTFE	PTFE
	Thickness (μm)	150	40	30	48	100	42	50
	SSE:Binder ratio	40:00.2	99.5:0.5	99.8:0.2	100:0.4	100:1	99.8:0.2	100:0.1
	Areal Mass (mg cm⁻²)	24.6	8.4	4.92	7.872	20.4	6.56	8.2
Cathode	Material	NCM 9055	LCO	NCM 523	Thiuram sulfide	NCM 523	NCM 622	NCM 82
	SSE	Li ₆ PS ₅ Cl	Li ₃ InCl ₆	Li _{5.4} PS _{4.4} Cl _{1.6}	Li ₆ PS ₅ Cl	Li ₁₀ GeP ₂ S ₁₂	Li ₆ PS ₅ Cl	Li ₆ PS ₅ Cl
	Binder	PTFE	PTFE	PTFE	PTFE	-	PTFE	PTFE
	Thickness (μm)	86	86.5	66.1	88.0	34.4	28.2	120
	AM:SSE:C:Binder ratio	85:13:2:0.3	85:15:0:0.5	70:30:2:1	33.3:16.7:50:0.8	65:25:10:0	60:40:3:0.5	80:17:3:0.5
	Areal capacity (mAh cm⁻²)	2.9	3.00	1.86	1.20	0.96	0.64	5
Areal mass (mg cm⁻²)	17.1	25.3	17.1	6.1	9.2	6.9	31.4	
Anode	Material	Graphite	Graphite	Al ₂ O ₃ @Li	LiIn	GPE@Li	LiIn	Si
	SSE	Li ₆ PS ₅ Cl	Li ₆ PS ₅ Cl	-	-	-	-	-
	Binder	PTFE	PTFE	-	-	-	-	PVDF
	Thickness (μm)	83.2	107.9	80	277	100	100	15
	AM:SSE:C:Binder ratio	60:35:5:0.5	50:50:0:0.5	100:0:0:0	100:0:0:0	100:0:0:0	100:0:0:0	100:0:0:0.1
	Areal capacity (mAh cm⁻²)	3.77	3.9	-	-	-	-	7
	Areal mass (mg cm⁻²)	17	21.1	4.3	92.8	5.3	73.1	2.0
	N/P ratio	1.3	1.3	-	-	-	-	1.4
Cell	Al thickness (μm)	10	10	10	10	10	10	10
	Cu thickness (μm)	10	10	10	10	10	10	10
	Total areal mass (mg cm⁻²)	70.3	66.5	37.9	118.4	46.6	98.2	53.2
	Total thickness (μm)	339.2	254.4	196.1	433.0	254.4	190.2	205
	Chemistry	Gr NCM9055	Gr LCO	Li NCM523	LiIn S	Li NCM523	LiIn NCM622	Si NCM82
	Nominal cell voltage (V)	3.55	3.55	3.7	1.375	3.7	3.075	3.35
Performance	Cell format	Pouch	Pouch	Pellet/Coin	Pellet/Coin	Pellet/Coin	Pellet/Coin	Pouch
	Current density (mA cm⁻²)	0.7 (0.24 C)	0.3 (0.1 C)	0.085 (0.05 C)	0.4 (0.33 C)	0.192 (0.2 C)	0.64 (1 C)	0.5 (0.1 C)
	Capacity utilization (%)	62.5	88.8	84.6	95	86.3	72.8	82.6
	Cycle number	100	50	150	500	50	1000	30
	Capacity retention (%)	93.6	68.6	80.1	80.8	88.4	86.4	83.2
	Specific energy (Wh kg⁻¹)	146.3	160.2	181.1	13.9	76.2	20.0	310
	Energy density (Wh L⁻¹)	303.5	418.6	350.1	38.1	139.6	103.5	805

As the reviewer correctly pointed out, our co-rolled film does not have the thinnest SSE layer compared to other works. However, since the primary objective of reducing the thickness of SSE layer is to enhance the energy density, our work has demonstrated the highest specific energy and energy density compared to other published works (Supplementary Table 1 and Fig. 6f). This was made possible due to a much higher areal loading of cathode and suitable selection of high-capacity anode. We believe that realizing a high-energy density cell is necessary along with decreasing the thickness of SSE layer.

In addition, we would like to clarify that the novelty of this work is on the improvement of the processibility and mechanical stability of thin SSE layers with our new dry-process design. Although other works have demonstrated the feasibility of dry-processing thin SSE film, the practicality of film fabrication has never been considered. Using this co-rolling approach, a thin SSE layer could be easily fabricated with a robust and intimate SSE-cathode interface. The robust interface enabled a strong adhesion of two layers that improved its processibility and mechanical property, and the intimate solid-solid contacts between two layers improved cycling stability at reduced operation pressure. To our knowledge, this work is the first to consider both the processibility and low stack pressure performance of thin SSE layer in dry-processed ASSBs. Moreover, a fast line speed could be achieved for thin SSE layer fabrication compared to other works (Supplementary Fig. 11), demonstrating its potential capability of high-throughput fabrication. Therefore, we strongly believe that this work will provide an insight on the scalable, large-fabrication design of dry-processing ASSBs.

We have edited the manuscript as below (Page 7, Main text):

“With these optimized fabrication parameters, a fast line speed (4 m min^{-1}) could be realized in our laboratory roller machine to fabricate a thin SSE layer compared to other published works (Supplementary Fig. 11), demonstrating its potential capability of high-throughput fabrication.”

Supplementary Fig. 11 | (a) Photos of co-rolled film fabricated at line speed of 4 m min^{-1} . (b) Comparison of line speed of thin dry-processed SSE layer with other published works. Note that this work is the first to consider the fabrication speed in dry-processed SSE films.

4. As shown in Figure S2, the cathode in Figure S2c exhibited the most uniform surface, however it is not the optimal experimental condition. The author should give a detailed discussion. Besides, the cathode obtained from dry process is usually not particularly uniform. Has the author investigated the impact of different conditions on this issue?

Response: The reviewer raised an insightful concern. We would like to explain that the angle/intensity of light was slightly different for the samples with a uniform SSE layer (Supplementary Fig. 2a, b) and those with a non-uniform SSE layer (Supplementary Fig. 2c, d). Those with a uniform SSE layer were taken in brighter light condition to capture any color contrast on SSE layer, whereas those with non-uniform SSE layer were taken in normal light condition as color contrast on SSE layer was obvious. The same angle as the SSE layer was used when taking photos of the cathode side of each sample, resulting in different color contrasts of the cathode side as well. Moreover, a few white spots visible on the surface of cathode side is possibly due to some dust from tissues inside Ar-filled glovebox, which is difficult to entirely control as we prohibit use of any kind of solvents inside the glovebox to maintain the best chemically inert environment for sensitive chemicals such as SSEs.

Unlike SSE side, the uniformity of cathode side is hard to be determined by the color contrast of photos. To investigate the uniformity of cathode side with different fabrication conditions, we outlined a flow chart of experimental steps for studying different fabrication parameters and provided SEM images of cathode side of co-rolled films fabricated with different parameters (Fig. R1). As shown from the flow chart, all samples underwent the same mixing protocol and only different fabrication parameters (type of CAM, co-rolling temperature, or reduction thickness) were used to fabricate co-rolled films. From SEM images obtained with both SE and BSE modes, we have confirmed the absence of any noticeable non-uniformity on the cathode side with different fabrication parameters.

Fig. R1 | (a) Flow chart of experimental steps to study different parameters: CAM, Co-rolling temperature, reduction thickness. SEM images of cathode side with SE and BSE modes using parameters of (b) PC-NCM / 120 °C / 20 μm , (c) SC-NCM / 120 °C / 20 μm , (d) SC-NCM / 30 °C / 20 μm , (e) SC-NCM / 120 °C / 100 μm .

5. The cross-sectional EDS mapping of the interphase between SSE and cathode should be provided to investigate the element distribution.

Response: We appreciate the reviewer for pointing this out. It is important to confirm the SSE-cathode interphase of co-rolled film with EDS mapping. We have included the EDS results of SSE-cathode interface (Supplementary Fig. 12).

We have edited the manuscript as below (Page 7, Main text):

“Moreover, SEM images and energy dispersive X-ray spectroscopy (EDS) mapping of a co-rolled film showed a dense surface of LPSCl on SSE side (Fig. 2h), intimate coverage of SC-NCM with LPSCl on cathode side (Fig. 2i), and desired SSE-cathode interphase from a cross-section (Supplementary Fig. 12).”

Supplementary Fig. 12 | (a) SEM image and (b-d) EDS mapping of SSE-Cathode interface of co-rolled film after press.

6. The pouch cell utilizing free-standing SSE should be used as comparison.

Response: We appreciate the reviewer for the constructive comment. We have included pouch cell data utilizing freestanding SSE films in comparison with co-rolled film (Supplementary Fig. 31).

We have edited the manuscript as below (Page 14-15, Main text):

“In addition, a pouch cell was assembled with co-rolled film and Si anode (Fig. 6d) and cycled at room temperature and stack pressure of 5 MPa over 30 cycles (Fig. 6e). The pouch cell delivered higher discharge capacities and specific energies than the freestanding counterparts (Supplementary Fig. 31) and showed a high stack-level specific energy and energy density of 310 Wh kg^{-1} and 805 Wh L^{-1} , respectively (Supplementary Fig. 32).”

Supplementary Fig. 31 | (a) Comparison of cycling performance of pouch cells assembled with co-rolled film and freestanding films at 0.1C (20 mA g^{-1}) with activation at 0.05 C (10 mA g^{-1}). Voltage profiles of **(b)** co-rolled film and **(c)** freestanding films.

7. There are some typos and format errors should be corrected such as two R3 in table 3. The author should read through the manuscript carefully and fix the errors.

Response: We thank the reviewer for pointing out mistakes. The corresponding typos and errors have been corrected.

Reviewer #2

This article provided a novel co-rolling approach to dry-processing ASSBs that can achieve an intimate contact between composite cathode and solid electrolyte. This new strategy enabled the long cyclability (> 80% capacity retention after 500 cycles) at low stack pressure (2 MPa) due to the robust interface. Also, this work shows the probability of practical application of ASSB by achieving high-energy density ASSB coupled with Si anode (315 Wh kg⁻¹ and 840 Wh L⁻¹). However, current manuscript has to be revised because authors still need to provide clearer explanation for the several missing information. In my opinion, authors should address to several questions and comments for the publication in Nature Communications.

Response: We first thank the reviewer for the constructive and insightful comments which helped us to clarify our arguments and improved the quality of the manuscript. We hope our responses herein provide answers to these questions.

1. In figure 2b, SEM images showed cracked PC-NCM particles. Several articles fabricated PC-NCM based cell under 400 MPa shown in reference papers [1]-[3]. The cracking of PC-NCM is not detected at as-prepared electrode in these papers. Therefore, co-rolling process and high pressure (500 MPa) might induce the mechanical cracking. Could the authors consider decreasing the fabrication pressure? Additionally, PC-NCM is able to increase the utilization of cathode because large NCM and small SSE can increase the relative particle size (λ) according to the reference paper [4]. From the perspective of composite cathode design, PC-NCM might be more suitable than SC-NCM at typical fabrication pressure (~300 MPa). Could you provide SEM images of SC- and PC-NCM fabricated at ~300 MPa along with the corresponding voltage profile?

Here, these are 4 reference papers.

[1] Han, Yoonjae, et al. "Single-or poly-crystalline Ni-rich layered cathode, sulfide or halide solid electrolyte: which will be the winners for all-solid-state batteries?." *Advanced Energy Materials* 11.21 (2021): 2100126.

[2] Shi, Tan, et al. "Characterization of mechanical degradation in an all-solid-state battery cathode." *Journal of Materials Chemistry A* 8.34 (2020): 17399-17404.

[3] Payandeh, Seyedhosein, et al. "The effect of single versus polycrystalline cathode particles on all-solid-state battery performance." *Advanced Materials Interfaces* 10.3 (2023): 2201806.

[4] Shi, Tan, et al. "High active material loading in all-solid-state battery electrode via particle size optimization." *Advanced Energy Materials* 10.1 (2020): 1902881.

Response: The reviewer raised insightful points on the cracking and size of CAM particles, as well as fabrication pressures. Please allow us to explain our selection of the material and fabrication parameter.

First, regarding the cracking CAM particles, we used a relatively high CAM ratio of 80wt%. After carefully reviewing the provided reference papers, we noticed that relatively low CAM ratios at lower than 70wt% were used (68wt%⁵, 60wt%⁶, and 69wt%⁷). Contrarily, in the case where higher CAM ratio is used (80 and 90wt%)⁸, cracking of CAM was also observed for the as-prepared electrodes. We think that this is because the softness of SSE helps buffer local strains applied on CAM particles during fabrication pressurization. With increasing CAM ratios, contact points of CAM particles without SSE particles increase, leading to high localized strains and eventually cracking of the CAM particles. According to the reviewer's suggestion, we have compared large PC-NCM and small SC-NCM powder cathode composite fabricated at 300 and 500 MPa (Supplementary Figs. 8, 9). From the top-view SEM images with SE mode, at 300 MPa, insufficiently deformed SSE particles were observed for both PC-NCM and SC-NCM. At 500 MPa, SSE particles were further deformed and established better contacts, which indicates that fabrication pressure of 300 MPa was insufficient for our cathode composition to form intimate contacts. From BSE mode, cracking of PC-NCM particles was still observed in powder cathode composite, indicating that the cracking of CAM in co-rolled film was not related to the co-rolling process but more to the particle packing. Likewise, from the electrochemical results using these cathode composites, those pressed at 500 MPa showed lower overpotential, better rate capability, and lower impedance than those pressed at 300 MPa. Moreover, SC-NCM showed higher discharge capacities, better rate capability, and lower impedance than PC-NCM. Therefore, these experimental results led us to use SC-NCM instead of PC-NCM and fabrication pressure of 500 MPa. However, it is important to note that this work does not intend to compare and claim that SC-NCM is better than PC-NCM as their particle sizes are intrinsically different.

5. Han, Y. *et al.* Single- or Poly-Crystalline Ni-Rich Layered Cathode, Sulfide or Halide Solid Electrolyte: Which Will be the Winners for All-Solid-State Batteries? *Adv. Energy Mater.* **11**, 2100126 (2021).
6. Shi, T. *et al.* Characterization of mechanical degradation in an all-solid-state battery cathode. *J. Mater. Chem. A* **8**, 17399–17404 (2020).
7. Payandeh, S. *et al.* The Effect of Single versus Polycrystalline Cathode Particles on All-Solid-State Battery Performance. *Adv. Mater. Interfaces* **10**, 2201806 (2023).
8. Kim, J. *et al.* High-Performance All-Solid-State Batteries Enabled by Intimate Interfacial Contact Between the Cathode and Sulfide-Based Solid Electrolytes. *Adv. Funct. Mater.* **33**, 2211355 (2023).

Second, several experimental and simulation studies have shown the advantage of small particle size CAM, especially for low stack pressure operation. For example, small SC-NCM showed much higher capacity utilization and initial Coulombic efficiency (ICE) than large PC-NCM at low stack pressure of 2.5 MPa due to mitigated contact loss between CAM and SSE during the volume change of CAM⁹. Also, decreasing CAM particle size was also proposed as a strategy to increase the rate capability at low stack pressure from a simulation result¹⁰. Indeed, different electrode design aspects including tortuosity, cathode-SSE contacts, and relative particle sizes should be well considered and balanced for the optimal battery performance. Moreover, as the reviewer

pointed out, the large relative particle size (λ) is also crucial to ensure sufficient ionic percolation pathways within the cathode composite, therefore achieving desired CAM utilization¹¹. Because testing different CAM or SSE particle size is experimentally challenging, we tested different CAM ratio, another critical parameter raised by the provided reference, to verify the degree of CAM utilization of our cathode system. We have provided the electrochemical results with different CAM ratios of 60, 70, and 80wt% (Supplementary Fig. 24). The results showed that 80wt% CAM ratio still exhibited around 93% discharge capacity of 60wt% CAM ratio at 75 MPa. Given that the cathode-level specific energy can be increased by 33% from 60wt% to 80wt% CAM ratio, 7% capacity loss is still acceptable. Thus, we have concluded that our cathode system is reasonable in terms of CAM utilization even with a high CAM ratio of 80wt%.

9. Doerrer, C. *et al.* High Energy Density Single-Crystal NMC/Li₆PS₅Cl Cathodes for All-Solid-State Lithium-Metal Batteries. *ACS Appl. Mater. Interfaces* **13**, 37809–37815 (2021).
10. Naik, K. G., Jangid, M. K., Vishnugopi, B. S., Dasgupta, N. P. & Mukherjee, P. P. Interrogating the Role of Stack Pressure in Transport-Reaction Interaction in the Solid-State Battery Cathode. *Adv. Energy Mater.* (2024) doi:10.1002/aenm.202403360.
11. Shi, T. *et al.* High Active Material Loading in All-Solid-State Battery Electrode via Particle Size Optimization. *Adv. Energy Mater.* **10**, 1902881 (2020).

We have edited the manuscript as below (Page 6, Main text):

“The cracking of large PC-NCM was also confirmed in powder cathode composite at different fabrication pressures of 300 and 500 MPa (Supplementary Fig. 8). Since small SC-NCM also showed higher discharge capacities, better rate capability, and lower SSE-cathode resistance compared to large PC-NCM (Supplementary Fig. 9), small SC-NCM was used for further discussion.”

(Page 14, Main text):

“It is critical to reduce the stack pressure as low as possible for practical implementation of ASSBs², while the low stack pressure performance is even more aggravated with increasing CAM ratio in high loading electrodes (Supplementary Fig. 24).”

Supplementary Fig. 8 | SEM images of powder cathode composite with PC-NCM fabricated at (a-c) 300 MPa and (d-f) 500 MPa and SC-NCM fabricated at (g-i) 300 MPa and (j-l) 500 MPa.

Supplementary Fig. 9 | (a, d) Voltage profiles at 0.1 C (20 mA g^{-1}), (b, e) rate test at 0.1, 0.2, 0.3, 0.5, 1 C ($20, 40, 60, 100, 200 \text{ mA g}^{-1}$, respectively), and (c, f) EIS of PC-NCM and SC-NCM, respectively, fabricated at 300 and 500 MPa. SC-NCM fabricated at 500 MPa showed the lowest polarization, highest rate capability, and lowest impedance.

Supplementary Fig. 24 | (a-c) Voltage profiles at 0.1 C (20 mA g⁻¹) and (d-f) rate test at 0.1, 0.2, 0.3, 0.5, 1 C (20, 40, 60, 100, 200 mA g⁻¹, respectively) of different cathode active material ratios of 60, 70, and 80wt%, respectively, in cathode.

2. Considering that the capacity of cathode is 5 mAh cm⁻², it seems to be very thick. How thick is the co-rolled cathode? Moreover, the cohesion of the cathode might be weak due to its thickness even though a robust interface is formed between the cathode and SSE. Could you provide the peel test result specifically for the cathode compared to conventional slurry casting method?

Response: We appreciate the reviewer's comments. The cross-sectional SEM images of the cathode layer of co-rolled film are provided (Fig. R2). It was shown that the film thickness is around 180 μm before press and 120 μm after press. The thickness after press is comparable to that of dry-processed cathode with similar areal loading and CAM ratio (23.5 mg cm⁻², 80wt%) as previously reported¹².

12. Lee, D. *et al.* Shear force effect of the dry process on cathode contact coverage in all-solid-state batteries. *Nat. Commun.* **15**, 4763 (2024).

Fig. R2 | Cross-sectional SEM images of cathode layer of co-rolled film (a) before press and (b) after press.

As the reviewer requested, we conducted peel-off tests of slurry-cast cathode and co-rolled SSE-cathode (Fig. R3). On the initial trial, due to the poor adhesion of slurry-cast cathode with Al substrate and robust adhesion of SSE-cathode layer of co-rolled film, the cohesion of cathode layer alone could not be evaluated (Fig. R3 a, b). Instead, we conducted peel-off tests with detached slurry-cast cathode and dry-processed cathode, and they both showed detached cathode layers on the first trial (Fig. R3 c, d). However, a clear difference was observed after notching the electrode, where the slurry cast electrode cracked easily on the edge, and both dry-processed cathode and co-rolled SSE-cathode exhibited intact edges after notching (Fig. R3 e). Certainly, the dry-processed cathode has been reported to exhibit better cohesion than slurry-cast cathode due to the fibrillated binder structure as previously reported¹³.

13. Yao, W. *et al.* A 5 V-class cobalt-free battery cathode with high loading enabled by dry coating. *Energy Environ. Sci.* **16**, 1620–1630 (2023).

Fig. R3 | Comparison of slurry-cast cathode, dry-processed cathode, and co-rolled cathode. Peel-off tests of (a) slurry cast cathode on Al substrate, (b) co-rolled SSE-cathode, (c) detached slurry-cast cathode, and (d) dry-processed cathode. (e) Photos of electrodes after notching. The binder ratio was all fixed to 0.5wt%. Nitrile butadiene rubber (NBR) dissolved in toluene was used for slurry-cast cathode.

3. In my opinion, conventional electron blocking cell configuration may not be applicable for co-rolled film, as I understand the configuration is LiIn/SSE/co-rolled film (cathode/SSE)/SSE/LiIn. Therefore, it is hard to measure Li^+ transport of SSE of solely. Also, the measured value might be effective Li^+ conductivity in composite cathode. Could you show the detailed calculation process for determining ionic conductivity? Additionally, could you provide the ionic conductivity of each SSE independently?

Response: We appreciate the reviewer for pointing this out. We acknowledge the lack of experimental details on the characterization of Li^+ and e^- transport properties in SSE and cathode layers. Please allow us to clarify this.

As the reviewer mentioned, due to the intrinsically integrated structure (SSE-cathode) of co-rolled film, characterizing each layer independently was challenging. We have added the schematic of cell configurations used for each characterization (Supplementary Fig. 19). For Li^+ and e^- transports in SSE, SSE-cathode configuration was used. We assumed that relatively high e^- conductivity of cathode layer ($> 30 \text{ mS cm}^{-1}$) had a negligible impact on the bulk resistance of electrochemical impedance spectroscopy (EIS) and current measured of direct current polarization (DCP). For Li^+ transport in cathode, the conventional electron-blocking cell configuration was used, where partial powder SSE and SSE layer of co-rolled film were used. For e^- transport,

cathode layer was independently tested, where the SSE layer of co-rolled film was gently removed by peeling-off with tapes. For equal comparison, both co-rolled and freestanding films were tested in the same configuration for each test. As the purpose of these characterizations was to ensure the comparable transport properties of co-rolled film compared to freestanding films, we believe that the above assumptions do not critically affect their comparison.

As the reviewer suggested, we also have added the calculation process and edited the manuscript as below (Page 17, Main text):

“Li⁺ or e⁻ conductivities were calculated using the following equation: $\sigma = \frac{L}{R \times A}$ (1) where L is the thickness of SSE or cathode layer, R is the resistance obtained from EIS or DCP measurements, and A is the cell area. For EIS, R1 was used for Li⁺ transport in SSE, and sum of R2 and R3 was used for Li⁺ transport in cathode (Supplementary Fig. 20). For DCP, resistance was obtained by using the following equation: $R = \frac{V}{I}$ (2) where V is the applied potential bias, and I is the current measured.”

(Page 11-12, Main text)

“Since their transport properties are affected by not only the composition but also structure after fabrication⁵¹, co-rolled and freestanding films of the same compositions are fabricated and characterized with appropriate cell configurations (Fig 4b and Supplementary Fig. 19).”

Supplementary Fig. 19 | Cell configurations and electrochemical methods used for the characterization of (a) Li⁺ transport in SSE, (b) e⁻ transport in SSE, (c) Li⁺ transport in cathode, and (d) e⁻ transport in cathode. Li⁺ and e⁻ transports in SSE were conducted with SSE-cathode structure due to the intrinsically integrated structure of co-rolled film. The high e⁻ conductivity of

cathode layer ($> 30 \text{ mS cm}^{-1}$) was assumed to have negligible impact on the bulk impedance of EIS for Li^+ transport in SSE and the current measured of DCP for e^- transport in SSE. For Li^+ transport in cathode, powder SSE was used for freestanding films, and partial power SSE and SSE layer of co-rolled film were used for co-rolled film. The overall thickness of SSE layers was fixed. For e^- transport in cathode, SSE layer of co-rolled film was carefully separated from cathode layer by peeling-off with tapes.

4. In figure 4c, authors provided conductance of both SSE. Given that the thickness of free-standing SSE is 10 times thicker than co-rolled SSE while the area remains same, authors should compare the conductivity rather than conductance. The higher resistance of free-standing SSE could be due to the longer Li^+ transport length ($500 \mu\text{m}$) as the resistance is proportional to thickness, as shown in the equation below. Additionally, the unit of conductance should be expressed as mS or S, not mS cm^{-2} .

$$R = L / (\sigma \times A)$$

R is resistance, σ is conductivity, L is thickness, and A is area.

Response: We thank the reviewer for pointing out our mistake. We have corrected the unit of conductance in the manuscript (Fig. 4c, inset).

Regarding the use of conductance instead of conductivity, we used conductance because this value is what eventually affects the cell resistance, which is also observed from the internal resistance test (Fig. 4g). So, we still used conductance in the bar chart for better understanding of our results in the internal resistance test. Instead, we added the conductivity values of both SSE layers of co-rolled and freestanding films in the main text.

We have edited the manuscript as below (Page 12, Main text):

“A much higher ionic conductance of co-rolled film than that of freestanding films (164 vs. 20 mS, corresponding to ionic conductivity of $1.04 \text{ vs. } 1.29 \text{ mS cm}^{-1}$, respectively) was observed, which was attributed to a much thinner SSE layer (Fig. 4c, Supplementary Fig. 20a and Supplementary Table 2).”

Fig. 4 | Electrochemical characterizations of co-rolled film. **a** Schematic illustrating Li^+ and e^- transports in SSE and cathode layers, and cell properties affected by such transports. **b** Co-rolled film and freestanding films used for the characterization. **c-f** Evaluation of Li^+ and e^- transports in SSE and cathode layers of co-rolled film and freestanding films. **c** Li^+ transport in SSE layer. **d** e^- transport in SSE layer. **e** Li^+ transport in cathode layer. **f** e^- transport in cathode layer. **g-i** Evaluation of cell properties of cells assembled with co-rolled film and freestanding films. **g** Internal resistance. **h** Shelf-life. **i** Stack pressure effects by DRT analysis.

5. Page 12, line 35, authors demonstrated a high-energy density ASSB with the co-rolled film. Please show the calculation process of gravimetric and volumetric energy density of above cell.

Response: We appreciate the reviewer for pointing this out. In our initial version of the manuscript, we used entirely theoretical values to calculate the energy densities. We have updated them with the obtained value of specific energy and added the calculation process and our assumptions (Supplementary Fig. 32, Supplementary Note 1 and Supplementary Table 1).

Supplementary Fig. 32 | Specific energy and energy density measurement of an ASSB pouch cell assembled with co-rolled SSE-cathode and Si anode at 0.05 C (10 mA g⁻¹). The stack-level (including cathode, SSE, anode layers, and Al and Cu current collectors) specific energy and energy density are calculated to be 310 Wh kg⁻¹ and 805 Wh L⁻¹, respectively.

Supplementary Note 1 | Specific energy and energy density calculations of co-rolled film.

Specific energy (Wh kg⁻¹) and energy density (Wh L⁻¹) in Fig. 6f were calculated based on stack-level (including SSE, cathode, anode layers, and Al and Cu current collectors). The obtained cathode-level specific energy was converted to stack-level specific energy or energy density by using the areal mass and thickness values of each layer stated in Supplementary Table 1. Specifically, areal mass of 8.2, 31.4, 2.0, 2.7, 8.96 mg cm⁻² and thickness of 50, 120, 15, 10, 10 μm were used for SSE, cathode, anode, Al, and Cu, respectively. This corresponds to a total areal mass of 53.2 mg cm⁻² and thickness of 205 μm of a cell stack.

6. Page 15, line 25, Si electrode is composed of binder and silicon without SSE in contrast to LTO electrode. The charge transfer reaction mainly occurred at the anode surface near SSE. Also, Si electrode might suffer dynamic volume change to accommodate lithium from 5 mAh cm^{-2} cathode. What is the thickness of the Si electrode before and after cycling? Additionally, is there any possibility that SSE could penetrate into the anode during 500 MPa cold isostatic press?

Response: We thank the reviewer for the insightful concerns.

We have provided the cross-sectional SEM images of Si at different stages of lithiation (Fig. R4). As the reviewer mentioned, the Si undergoes a large volume change during cycling. The pristine μSi cast shows thickness of $18 \mu\text{m}$, expands to a $30 \mu\text{m}$ dense lithiated Si after lithiation, and shrinks to a $15 \mu\text{m}$ columnar delithiated Si. Although Si undergoes a large volume change, lithiated Si particles can deform and form contacts if sufficient and uniform pressures are applied^{1,2}. For example, we have recently reported that 99.9wt% Si anode could be cycled stably over hundreds of cycles at 5 MPa and room temperature³. Certainly, there are other challenges to further improve its cyclability and kinetics, but we would like to be clear that Si anode was used as a demonstration of high-energy density ASSBs as the focus of this work is on the fabrication of SSE and cathode layers.

1. Cao, Q. *et al.* Stacking pressure homogenizes the electrochemical lithiation reaction of silicon anode in solid-state batteries. *Energy Storage Mater.* **67**, 103246 (2024).
2. Nelson, D. L. *et al.* Fracture Dynamics in Silicon Anode Solid-State Batteries. *ACS Energy Lett.* 6085–6095 (2024) doi:10.1021/acenergylett.4c02800.
3. Chen, Y. *et al.* Enabling Uniform and Accurate Control of Cycling Pressure for All-Solid-State Batteries. *Adv. Energy Mater.* **14**, 2304327 (2024).

Fig. R4 | Cross-sectional SEM images of (a) pristine μSi cast on Cu current collector, (b) lithiated dense Si, and (c) delithiated columnar Si.

We also checked the cross-sections of Si and SSE interface after cold isostatic press of 500 MPa (Fig. R5). It was confirmed that neither Si nor SSE has penetrated to a different layer after cold isostatic press from SEM image and EDS mappings.

Fig. R5 | (a) Cross-sectional SEM image of Si-SSE interface after cold isostatic press of 500 MPa, and EDS mapping of (b) Si and (c) S.

7. In Supplementary Fig. 14, authors proposed the interfacial fibrillation mechanism during co-rolling process. The co-rolled process can form the robust interface between cathode and SSE shown in figure 3 d, g. The improved electrochemical performance shows the importance of intimate contact. However, it is insufficient to verify the proposed mechanism. Therefore, could you provide the data that can verify your mechanism? It could intensify the meaning of the robust interface for practical application of ASSBs.

Response: We thank the reviewer for the sincere feedback. We acknowledge that our initial version of the manuscript might have lacked sufficient detail on mechanism of interfacial fibrillation and intimate interfacial contact, as well as how this interface evolved and maintained. Please allow us to clarify these aspects.

First, we wish to highlight the intrinsic difference between the SSE-cathode interface structures of co-rolled film and freestanding films. Unlike single-layer fabrication (*i.e.*, freestanding SSE and cathode films), dual-layer fabrication (*i.e.*, co-rolled SSE-cathode film) involves the excessive degree of shear applied to the SSE-cathode interface during co-rolling (Fig. 3j). This shearing on the interface leads to two effects: (i) binder fibrillation and (ii) contact formation on the interface. (i) The mechanism of the binder fibrillation at the interface was supported by the progression of the binder fibrillation in the initial and final stages of co-rolling (Supplementary Fig. 18). From SEM images, the binders present on the surface of SSE and cathode layers make contacts in the initial stage, and after excessive shearing is applied, binders are fibrillated across the interface in the final stage. (ii) Intimate particle-to-particle contact formation during co-rolling led to intimate SSE-cathode interface (fused interface) compared to less intimate SSE-cathode interface (heterogeneous interface) in freestanding films (Figure 3k, l). This fused interface is beneficial under low stack pressure cycling (*e.g.*, 2 MPa) where voids and cracks are formed and maintained during cycling. Thus, we think that these are strong evidence of intrinsically different interface structure of co-rolled film compared to the freestanding films.

Second, to investigate interface evolution during charge and discharge, we performed in-situ EIS-DRT analysis on co-rolled film and freestanding films at 2 MPa (Supplementary Fig. 23). As

shown from the contour mappings, interface evolution around 10^{-4} s (indicating cathode-SSE resistance) of the time constant is observed during charge process. The cell with freestanding films shows much higher resistance of this peak than that with co-rolled film at the end of the charge, and this peak is mostly maintained during the whole discharge process. The result is also consistent with the ex-situ measurement at 2 MPa (Fig. 3i). This distinct behavior of interface evolution provides strong evidence that the intimate SSE-cathode interface in co-rolled film plays a crucial role at low stack pressure.

Third, pressurization control is also crucial for low stack pressure cycling as the dynamic volume change occurs during charge and discharge. We have monitored the pressure change using LTO || NCM system at stack pressures of 75 and 2 MPa (Supplementary Fig. 25). Using fixed gap setup, the pressure decreases of 1.5 and 0.3 MPa were observed for 75 and 2 MPa, respectively. This change corresponds to a 2% decrease for 75 MPa but 15% decrease for 2 MPa. Therefore, constant pressure setup was used for 2 MPa cycling to maintain consistent pressurization during long-term cycling.

We have edited the manuscript as below (Page 9-10, Main text):

“The primary difference between co-rolled film and freestanding films from a fabrication viewpoint is the degree of shear applied on the SSE-cathode interface. As the shear force has been reported to induce (i) fibrillation of binder³⁶ and (ii) contact formation of SSE particles³⁷ in the single-layer fabrication, such effects are extended to the interface in the dual-layer fabrication (Fig. 3j). For freestanding films, with no shear applied on the interface, (i) interface adhesion is determined by binders present on the surface of SSE and cathode layers, and (ii) interface contact is determined by layer-to-layer contacts made during stacking. On the other hand, for co-rolled film, with excessive shear applied on the interface during the feed reduction step, complex mechanical dynamics of the particles and binders occur at the SSE-cathode interface. This shear applied on the interface induces (i) movement of binder contacts yielding binder fibrillation across the interface (Supplementary Fig. 18) and (ii) formation of intimate particle-to-particle contacts at the interface yielding a fused interface for co-rolling film, after press, compared to a heterogenous interface for freestanding films (Fig. 3k, j). Therefore, co-rolling dry-process not only improves the processability of thin SSE layer, but also constructs the robust and intimate SSE-cathode interface structure, in both film and cell states, compared to the freestanding counterparts.”

(Page 12, Main text)

“Consequently, while cells with both co-rolled and freestanding films showed comparable discharge capacities (~ 191 mAh g^{-1}) at 75 MPa (Supplementary Fig. 22b), a cell with co-rolled film showed a much higher discharge capacity than that with freestanding films (177 vs. 141 mAh g^{-1}) at 2 MPa (Supplementary Fig. 22c). The in-situ EIS-DRT analysis further confirmed the

different behavior of their interface evolution, where the cell assembled with co-rolled film maintained stable interface impedance during charge/discharge at 2 MPa but that with freestanding films showed a larger increase in impedance (Supplementary Fig. 23).”

(Page 14, Main text)

“Here, $\text{Li}_4\text{Ti}_5\text{O}_{12}$ (LTO) was used as a counter electrode (i) to isolate the pressure effects on cathode side owing to the low volume change ($\sim 0.2\%$) of LTO during charging and discharging^{53,54} and (ii) to evaluate the performance of SSE-cathode structure independently by preventing the reduction of PTFE binder (< 1 V vs. Li/Li^+) within SSE layer³⁹. To maintain the pressure during dynamic volume change of the cathode, the constant pressure setup was used for 2 MPa cycling (Supplementary Fig. 25).”

Supplementary Fig. 18 | (a) A proposed mechanism of interfacial fibrillation during co-rolling process. (i) Several different contacts can be formed at SSE-cathode interface (dashed line): (1) Binder in SSE layer-to-binder in cathode layer, (2) SSE particle in SSE layer-to-binder in cathode layer, and (3) binder in SSE layer-to-VGCF in cathode layer. (ii) Shearing induces movement of materials at the interface as well as stress on the binder. (iii) Due to the movement, binders are stretched and fibrillated across the interface. As a result, the initial contacts (1-3) have moved across the interface. (iv) New contacts have formed from the interfacial fibrillation of binder. (v) This process is repeated with every step of thickness reduction. SEM images of SSE-cathode interface during (b) initial stage of co-rolling showing binder contacts at the interface and (c) final stage of co-rolling showing binder fibrillation across the interface.

Fig. 3 | Physical and mechanical characterizations of co-rolled film and interface. **a** Flexibility test of co-rolled film. **b** Tensile strength measurement of SSE and cathode freestanding films and co-rolled film. **c** Comparison of SSE binder content and SSE thickness with other published works on dry-processed SSE film^{35,46–50}. **d** Peel-off test of freestanding films after lamination. **e** Side-view SEM image of freestanding films. **f** Schematic of freestanding films showing partially

attached SSE-cathode interface and limited interface adhesion. **g** Peel-off test of co-rolled film. **h** Side-view SEM image of co-rolled film. **i** Schematic of co-rolled film showing binder-VGCF network and improved interface adhesion. **j** Schematic illustrating shearing effect on the interface: binder fibrillation and contact formation. **k** SEM images and **l** schematics of freestanding and co-rolled films showing heterogenous interface and fused interface, respectively, after press. Yellow spheres represent SSE and gray spheres represent CAM.

Supplementary Fig. 23 | In-situ observation of interface evolution of co-rolled films and freestanding films at 2 MPa in a half-cell configuration using LiIn as the counter electrode. EIS measurement and DRT analysis, respectively, of **(a, b)** discharge process and **(c, d)** charge process of co-rolled film. EIS measurement and DRT analysis, respectively, of **(e, f)** discharge process and **(g, h)** charge process of freestanding films.

Supplementary Fig. 25 | Cell design and pressurization methods used to evaluate long-term cycling stability of SSE-cathode structures. **(a)** A schematic of LTO || NCM full-cell with their volume change during charging. Photos of **(b)** fixed gap setup and **(c)** constant pressure setup. Monitored pressure changes during cell operation of **(d)** 75 MPa – fixed gap, **(e)** 2 MPa – fixed gap, and **(f)** 2 MPa – constant pressure setups. For long-term cycling stability test at different stack pressures, the fixed gap setup was used for 75 MPa, and the constant pressure setup was used for 2 MPa. Note that the constant pressure setup could not be used for 75 MPa due to the lack of suitable springs to sustain such high pressure.

Reviewer #3

Dry battery technology is an emerging field that is garnering significant attention from both academia and industry due to its potential impact on the future of the all-solid-state battery industry. However, the high operational pressure required for all-solid-state batteries remains a significant barrier to their widespread adoption. This paper proposed a co-rolling dry-process technique to reduce of battery operation pressure and improve the mechanical stability of solid-state electrolyte film, from the film fabrication through to cell operation along with cathode interfaces. The thickness of solid electrolyte layer was successfully reduced to 50 μm , and a high loading cathode layer (5 mAh cm^{-2}) was fabricated without observed cracks or tears. The resulting all-solid-state batteries exhibited an enhanced cycle stability of 500 cycles under 2 MPa pressure. Overall, this research offers valuable insights in reducing battery operation pressure through modified dry processes. The manuscript is well-organized and will attract broad interests in the field of all-solid-state lithium batteries. However, after a thorough review, I have identified several issues that need to be addressed before it is suitable for publication in Nature Communications. These issues are outlined below:

Response: We first thank the reviewer for the constructive feedback. We hope our responses herein provide answers to these questions.

1. While I am convinced that a low external pressure (2 MPa) is sufficient during the initial stages of battery cycling, the battery undergoes significant volume changes during continuous charging and discharging. These changes will lead to the loss of contact between solid particles due to increased internal pressures (generally >2 MPa). The authors should provide additional post-mortem analyses and discuss how the co-rolling dry-process technique helps maintain a conformal interface during long-term cycling.

Response: We appreciate the reviewer for the insightful comment, which also echoed with Reviewer #2. We acknowledge that our first version of the manuscript might have lacked sufficient detail on interfacial contacts and their stability during charge and discharge process, as well as in-situ characterization data. Please allow us to clarify these aspects (also kindly refer to our response in addressing Reviewer #2).

First, we wish to highlight the intrinsic difference between the SSE-cathode interface structures of co-rolled film and freestanding films. Unlike single-layer fabrication (*i.e.*, freestanding SSE and cathode films), dual-layer fabrication (*i.e.*, co-rolled SSE-cathode film) involves the excessive degree of shear applied to the SSE-cathode interface during co-rolling process (Fig. 3j). This shearing creates intimate particle-to-particle contacts that yield, after fabrication pressure, intimate SSE-cathode interface (fused interface) in co-rolled film compared to less intimate SSE-cathode interface (heterogenous interface) in freestanding films (Figure 3k, l). This fused interface is beneficial under low stack pressure cycling (*e.g.*, 2 MPa) where voids and cracks are formed and maintained during cycling.

Second, we performed in-situ EIS-DRT analysis on co-rolled film and freestanding films at 2 MPa to investigate interface evolution during charge and discharge (Supplementary Fig. 23). As shown from the contour mappings, interface evolution around 10^{-4} s (indicating cathode-SSE resistance) of the time constant is observed during charge process. The cell with freestanding films shows much higher resistance of this peak than that with co-rolled film at the end of the charge, and this peak is mostly maintained during the whole discharge process. The result is also consistent with the ex-situ measurement at 2 MPa (Fig. 3i). This distinct behavior of interface evolution provides strong evidence that the intimate SSE-cathode interface in co-rolled film plays a crucial role at low stack pressure. Furthermore, the post-mortem analysis of the interface after long-term cycling further explains the different degree of interfacial void formation between co-roll film and freestanding films (Fig. 5e, f).

Third, pressurization control is also crucial for low stack pressure cycling. As the reviewer pointed out, the dynamic volume change occurs during charge/discharge. We have monitored the pressure change using LTO || NCM system at stack pressures of 75 and 2 MPa (Supplementary Fig. 25). Using fixed gap setup, the pressure decreases of 1.5 MPa and 0.3 MPa were observed for 75 MPa and 2 MPa, respectively. This change corresponds to a 2% decrease for 75 MPa but 15% decrease for 2 MPa. Therefore, constant pressure setup was used for 2 MPa cycling to maintain consistent pressurization during long-term cycling.

In summary, the intimate interface could be maintained for co-rolled film at low stack pressure (1) due to the intimate and fused SSE-cathode interface, (2) which exhibited much lower resistance than freestanding films from in-situ EIS-DRT analysis due to less void formation at the interface, and (3) employing the constant pressurization setup for long-term cycling.

We have edited the manuscript as below (Page 9-10, Main text):

“The primary difference between co-rolled film and freestanding films from a fabrication viewpoint is the degree of shear applied on the SSE-cathode interface. As the shear force has been reported to induce (i) fibrillation of binder³⁶ and (ii) contact formation of SSE particles³⁷ in the single-layer fabrication, such effects are extended to the interface in the dual-layer fabrication (Fig. 3j). For freestanding films, with no shear applied on the interface, (i) interface adhesion is determined by binders present on the surface of SSE and cathode layers, and (ii) interface contact is determined by layer-to-layer contacts made during stacking. On the other hand, for co-rolled film, with excessive shear applied on the interface during the feed reduction step, complex mechanical dynamics of the particles and binders occur at the SSE-cathode interface. This shear applied on the interface induces (i) movement of binder contacts yielding binder fibrillation across the interface (Supplementary Fig. 18) and (ii) formation of intimate particle-to-particle contacts at the interface yielding a fused interface for co-rolling film, after press, compared to a heterogenous interface for freestanding films (Fig. 3k, j). Therefore, co-rolling dry-process not only improves

the processability of thin SSE layer, but also constructs the robust and intimate SSE-cathode interface structure, in both film and cell states, compared to the freestanding counterparts.”

(Page 12, Main text)

“Consequently, while cells with both co-rolled and freestanding films showed comparable discharge capacities ($\sim 191 \text{ mAh g}^{-1}$) at 75 MPa (Supplementary Fig. 22b), a cell with co-rolled film showed a much higher discharge capacity than that with freestanding films (177 vs. 141 mAh g^{-1}) at 2 MPa (Supplementary Fig. 22c). The in-situ EIS-DRT analysis further confirmed the different behavior of their interface evolution, where the cell assembled with co-rolled film maintained stable interface impedance during charge/discharge at 2 MPa but that with freestanding films showed a larger increase in impedance (Supplementary Fig. 23).”

(Page 14, Main text)

“Here, $\text{Li}_4\text{Ti}_5\text{O}_{12}$ (LTO) was used as a counter electrode (i) to isolate the pressure effects on cathode side owing to the low volume change ($\sim 0.2\%$) of LTO during charging and discharging^{53,54} and (ii) to evaluate the performance of SSE-cathode structure independently by preventing the reduction of PTFE binder ($< 1 \text{ V vs. Li/Li}^+$) within SSE layer³⁹. To maintain the pressure during dynamic volume change of the cathode, the constant pressure setup was used for 2 MPa cycling (Supplementary Fig. 25).”

Fig. 3 | Physical and mechanical characterizations of co-rolled film and interface. **a** Flexibility test of co-rolled film. **b** Tensile strength measurement of SSE and cathode freestanding films and co-rolled film. **c** Comparison of SSE binder content and SSE thickness with other published works on dry-processed SSE film^{35,46–50}. **d** Peel-off test of freestanding films after lamination. **e** Side-view SEM image of freestanding films. **f** Schematic of freestanding films showing partially

attached SSE-cathode interface and limited interface adhesion. **g** Peel-off test of co-rolled film. **h** Side-view SEM image of co-rolled film. **i** Schematic of co-rolled film showing binder-VGCF network and improved interface adhesion. **j** Schematic illustrating shearing effect on the interface: binder fibrillation and contact formation. **k** SEM images and **l** schematics of freestanding and co-rolled films showing heterogenous interface and fused interface, respectively, after press. Yellow spheres represent SSE and gray spheres represent CAM.

Supplementary Fig. 23 | In-situ observation of interface evolution of co-rolled films and freestanding films at 2 MPa in a half-cell configuration using LiIn as the counter electrode. EIS measurement and DRT analysis, respectively, of **(a, b)** discharge process and **(c, d)** charge process of co-rolled film. EIS measurement and DRT analysis, respectively, of **(e, f)** discharge process and **(g, h)** charge process of freestanding films.

Supplementary Fig. 25 | Cell design and pressurization methods used to evaluate long-term cycling stability of SSE-cathode structures. **(a)** A schematic of LTO || NCM full-cell with their volume change during charging. Photos of **(b)** fixed gap setup and **(c)** constant pressure setup. Monitored pressure changes during cell operation of **(d)** 75 MPa – fixed gap, **(e)** 2 MPa – fixed gap, and **(f)** 2 MPa – constant pressure setups. For long-term cycling stability test at different stack pressures, the fixed gap setup was used for 75 MPa, and the constant pressure setup was used for 2 MPa. Note that the constant pressure setup could not be used for 75 MPa due to the lack of suitable springs to sustain such high pressure.

2. The authors mentioned that this co-rolling dry-process utilized very low binder content ($< 0.1\text{wt}\%$) while still achieving a thin SSE layer (page 8). However, there is no evidences or discussions of the interactions between the binders and active material particles. How about the interactions and how they change under varying pressures?

Response: The reviewer raised an interesting point. As we have discussed in the previous question, the intimate interface developed during co-rolling process is mostly responsible for the improved cycle performance at low stack pressure.

The interaction between binder and active material is another interesting aspect to investigate. We would first like to make it clear that $< 0.1\text{wt}\%$ of binder was used in SSE layer, and $0.5\text{wt}\%$ of binder was used in the cathode. For clarity, we have modified the y label in Fig. 3c (Binder ratio to SSE binder ratio). For SSE layer, we wanted to minimize the binder ratio to (i) demonstrate the minimal binder reliance of thin SSE layer using co-rolling approach and (ii) mitigate the conductivity loss of SSE from the binder and well as the reactivity of PTFE on the anode side¹⁴. Similarly, for the cathode layer, binder content had to be carefully controlled to minimize its effects on ionic and electronic conductivities of cathode layer. We have provided voltage profile and

electrochemical characterization results using binder ratio of 0.5, 2, 5wt% in cathode composite (Supplementary Fig. 5). The voltage profile showed that CAM utilization significantly decreased for 2 and 5wt% even under high stack pressure of 75 MPa. This was due to a decrease in both ionic and electronic conductivities in cathode composite with higher binder ratio as supported from EIS and DCP results. Therefore, we utilized 0.5wt% of binder ratio in cathode composite.

14. Lee, D. J. *et al.* Physio-Electrochemically Durable Dry-Processed Solid-State Electrolyte Films for All-Solid-State Batteries. *Adv. Funct. Mater.* **33**, 2301341 (2023).

We have edited our manuscript as below (Page 6, Main text):

“The active material ratio in cathode was fixed to 80wt% for high energy density, and the binder ratio of 0.5wt% was used to minimize the hinderance of Li^+ and e^- transports in the cathode layer (Supplementary Fig. 5).”

Supplementary Fig. 5 | (a) Voltage profiles at 0.1 C (20 mA g^{-1}) of cathode films with different ratios of PTFE binder. **(b)** Li^+ transport and **(c)** e^- transport properties obtained from electron-blocking and electron-nonblocking cell configurations, respectively. The ionic conductivities were calculated to be 0.069, 0.024, 0.007 mS cm^{-1} , and electronic conductivities were calculated to be 34, 4.5, 0.011 mS cm^{-1} for PTFE ratios of 0.5, 2, 5wt%, respectively. The weight ratio of CAM:SSE:VGCF was fixed to 80:17:3.

3. One key limitation of dry battery technology is the production speed. The authors should address the production speed achieved with this technique. Moreover, although the authors cite examples of the dry method from academic papers, they should also include references to its application in commercial battery manufacturing, such as patents, to demonstrate its industrial relevance.

Response: We thank the reviewer for mentioning this. As the reviewer suggested, we tried the fastest line speed (4 m min^{-1}) achievable with our equipment (Supplementary Fig. 11). Although

the value itself is much lower than the industry relevant values ($> \text{tens of m min}^{-1}$), we would like to highlight that this is the first reported value compared to other literatures.

In addition, as the reviewer recommended, we have added patents to further demonstrate the industrial relevance of dry-process.

We have edited our manuscript as below (Page 2, Main text):

“The dry-process is a promising option to eliminate solvents in the fabrication process^{29,30} and is extensively investigated even in the industry for commercial battery manufacturing^{31–34}.”

31. Zhong, L., Xi, X., Mitchell, P. & Zou, B. Dry particle based capacitor and methods of making same. US7352558B2 (2008).

32. Tschöcke, S. *et al.* Process for producing a dry film and dry film and dry film coated substrate. DE102017208220A1 (2018).

33. Wang, Y., Zheng, Z., Ludwig, B. & Pan, H. Dry powder based electrode additive manufacturing. US10547044B2 (2020).

34. Kim, M.-C. Dry electrode manufacturing method and dry electrode manufacturing system. WO2024144216A1 (2024).

(Page 7, Main text):

“With these optimized fabrication parameters, a fast line speed (4 m min^{-1}) could be realized in our laboratory roller machine to fabricate a thin SSE layer compared to other published works (Supplementary Fig. 11), demonstrating its potential capability of high-throughput fabrication.”

Supplementary Fig. 11 | (a) Photos of co-rolled film fabricated at line speed of 4 m min^{-1} . **(b)** Comparison of line speed of thin dry-processed SSE layer with other published works. Note that this work is the first to consider the fabrication speed in dry-processed SSE films.

4. If the authors claim an energy density of 315 Wh kg^{-1} and 840 Wh L^{-1} (Figure 6e), they should provide the actual overall weight and discharge capacity of the pouch cell. Additionally, the authors should clarify why the EIS profiles of SC/PC-NCM samples exhibit different shapes in Figure S7b. The XPS peaks are also supposed to be identified in Figure S10.

Response: We thank the reviewer for pointing these out.

For energy density, in our initial version of the manuscript, we used entirely theoretical values to calculate the energy densities. We have updated them with the obtained value of specific energy and added the calculation process (Supplementary Fig. 32, Supplementary Note 1 and Supplementary Table 1). Also, to avoid confusion, we have changed “energy density” to “stack-level energy density”.

Supplementary Fig. 32 | Specific energy and energy density measurement of an ASSB pouch cell assembled with co-rolled SSE-cathode and Si anode at 0.05 C (10 mA g^{-1}). The stack-level (including cathode, SSE, anode layers, and Al and Cu current collectors) specific energy and energy density are calculated to be 310 Wh kg^{-1} and 805 Wh L^{-1} , respectively.

Supplementary Note 1 | Specific energy and energy density calculations of co-rolled film.

Specific energy (Wh kg^{-1}) and energy density (Wh L^{-1}) in Fig. 6f were calculated based on stack-level (including SSE, cathode, anode layers, and Al and Cu current collectors). The obtained cathode-level specific energy was converted to stack-level specific energy or energy density by using the areal mass and thickness values of each layer stated in Supplementary Table 1. Specifically, areal mass of 8.2, 31.4, 2.0, 2.7, 8.96 mg cm^{-2} and thickness of 50, 120, 15, 10, 10 μm were used for SSE, cathode, anode, Al, and Cu, respectively. This corresponds to a total areal mass of 53.2 mg cm^{-2} and thickness of 205 μm of a cell stack.

For the EIS results comparing PC-NCM and SC-NCM, the different shape of EIS profile in the initial version of the manuscript is due to the different degree of delithiation/lithiation of both anode (Si) and cathode (PC- or SC-NCM) comprising a full-cell. Because two cathodes exhibited different ICE and the measurement was taken at a discharged state, different lithiation state of Si and PC- or SC-NCM might have showed different shapes of EIS profile. For clarity, we have replaced it by taking EIS at 50% SOC and using Li-In as the anode to construct a half-cell (Supplementary Fig. 9).

Supplementary Fig. 9 | (a, d) Voltage profiles at 0.1 C (20 mA g^{-1}), (b, e) rate test at 0.1, 0.2, 0.3, 0.5, 1 C (20, 40, 60, 100, 200 mA g^{-1} , respectively), and (c, f) EIS of PC-NCM and SC-NCM, respectively, fabricated at 300 and 500 MPa. SC-NCM fabricated at 500 MPa showed the lowest polarization, highest rate capability, and lowest impedance.

For XPS results, we have identified the peaks of S 2p and Cl 2p (Supplementary Fig. 14).

Supplementary Fig. 14 | (a) S 2p, (b) Cl 2p, (c) Ni 2p XPS of SSE side and cathode side of controlled film.

5. Binder selection is critical for dry electrodes. It would be beneficial for the authors to test and discuss suitable binders for the co-rolling dry-process technique, including the pressures applied and their compatibility with different solid-state battery systems. This information would greatly assist readers in consulting related research.

Response:

We appreciate the reviewer's constructive and important suggestion.

It is important to provide more binder options to replace PTFE used in this work to not only stabilize the anode interface, but also show compatibility in other solid-state battery systems (*e.g.*, polymer- or oxide-based). However, we would like to express our greatest regret that we do not have much experience on other solid-state battery systems, and we focused only on the sulfide-based solid-state battery system. Instead, after a comprehensive literature survey, we have provided a summary table of different binders used in dry-process (Supplementary Table 4).

We have assessed the compatibility of each binder with our co-rolling dry-process based on the criteria of binder property, binding type, and fabrication method. Due to its similar aspects, we would like to carefully suggest a potential compatibility of ethylene-vinyl acetate (EVA) to replace PTFE. Since the comparison and testing of different binders would require careful material selection and optimization in the fabrication process, we believe that such study unfortunately falls beyond the scope of this work. Therefore, we would be happy to investigate in this research direction and provide interesting results in our future work.

We have edited our manuscript as below (Page 15, Main text):

“It should be noted that further cycling induced severe current leakage due to reduction of PTFE, which requires future works on stabilizing the anode interface to prevent the reduction or investigating different binders compatible with co-rolling dry-process (Supplementary Table 4).”

Supplementary Table 4 | Summary of binders used in dry-process and potential compatibility with co-rolling dry-process based on the criteria of binder property, binding type, and fabrication method.

Binder	Polytetrafluoro ethylene (PTFE)	Polyvinylidene fluoride (PVDF)	Paraffin	Ethylene-vinyl acetate (EVA)	Hydrogenated nitrile butadiene rubber (HNBR)	Styrene-butadiene rubber (SBR)
Binder property	Thermoplastic	Thermoplastic	Thermoplastic	Thermoplastic	Thermoset elastomer	Thermoset elastomer
Binding type	Fibrillated	Melted	N/A	Fibrillated	N/A	N/A
Fabrication method	Roll/Shear	Dry spray, Mold-press	Roll/Shear	Roll/Shear	Roll/Shear	Roll/Shear
Compatibility with co-rolling	Compatible (This work)	Potentially not compatible	Potentially less compatible	Potentially compatible	Potentially less compatible	Potentially less compatible
Reference	S1, S2	S3, S4	S5	S6	S7	S8

References:

S1. Zhang, Z., Wu, L., Zhou, D., Weng, W. & Yao, X. Flexible Sulfide Electrolyte Thin Membrane with Ultrahigh Ionic Conductivity for All-Solid-State Lithium Batteries. *Nano Lett.* **21**, 5233–5239 (2021).

S2. Hippauf, F. *et al.* Overcoming binder limitations of sheet-type solid-state cathodes using a solvent-free dry-film approach. *Energy Storage Mater.* **21**, 390–398 (2019).

S3. Ludwig, B., Zheng, Z., Shou, W., Wang, Y. & Pan, H. Solvent-Free Manufacturing of Electrodes for Lithium-ion Batteries. *Sci. Rep.* **6**, 23150 (2016).

S4. Ryu, M., Hong, Y.-K., Lee, S.-Y. & Park, J. H. Ultrahigh loading dry-process for solvent-free lithium-ion battery electrode fabrication. *Nat. Commun.* **14**, 1316 (2023).

S5. Kim, M. K. *et al.* Fluorine-Free Paraffin Binder-Based Dry Thick Electrodes Toward Sustainable and Efficient Battery Manufacturing. (2024) doi:10.2139/ssrn.4951163.

S6. Zhu, X., Jiang, W., Wang, L. & Lu, J. Constructing Resilient Cross-Linked Network Toward Stable All-Solid-State Lithium-Sulfur Batteries. *Adv. Energy Mater.* **14**, (2024).

S7. Khakani, S. E. *et al.* Melt-processed electrode for lithium ion battery. *J. Power Sources* **454**, 227884 (2020).

S8. Li, Y. *et al.* Long-Life Sulfide All-Solid-State Battery Enabled by Substrate-Modulated Dry-Process Binder. *Adv. Energy Mater.* **12**, 2201732 (2022).

Reviewer #4

The authors present an insightful approach of simultaneously rolling the cathode and SSE electrolyte films of dry processed cells. The approach has yielded significant improvement in the cathode physical and mechanical features which this reviewer has struggled with. The process excellently lends itself to roll-to-roll processing and may make a significant difference in this field.

- The device is well characterized with a good combination of imaging and advanced electrochemical diagnostics.

- After careful review, I find the manuscript to be well-written with excellent flow and grammar. The manuscript presents sufficient evidence for all claims and sufficient characterization overall.

Critically inspecting the manuscript for missing information, this reviewer cannot find major flaws for the authors to address. Therefore, this reviewer believes the manuscript can be published as-is.

Response: We thank the reviewer for appreciating our work. We hope that our work provides an insight in the field of large-scale fabrication of all-solid-state batteries.

We are grateful to the reviewers for their valuable comments and feedback to improve our manuscript. We believe that the comments and questions raised by the reviewers have helped us clarify our arguments and show the novelty of our work. We respond to each question raised by the reviewers in blue, and highlight the changes made to our manuscript in yellow.

Reviewer #1

The authors revised the manuscript carefully, I recommend acceptance of this revised manuscript.

Response: We are grateful for the reviewer's constructive comments and approval for publication.

Reviewer #2

The revised manuscript provides a more detailed explanation of the formation of the fused interface between SSE and cathode, emphasizing its critical role in achieving a robust interface. In-situ EIS analysis under the low stack pressure cycling shows the low resistance from the SSE-cathode interface. However, this revised manuscript also has several ambiguous explanations. Therefore, authors need to address to questions in the manuscript.

Response: We thank the reviewer for the constructive comments. We hope our responses herein answer the questions raised by the reviewer.

1. Authors assumed that the cathode layer exhibits relatively high electronic conductivity ($> 30 \text{ mS cm}^{-1}$). As shown in figure 4f, the electronic conductivity in cathode is about 30 mS cm^{-1} , which is 10^8 times larger than that in SSE in figure 4d. This suggests that the electronic conductivity has a negligible impact on the bulk resistance observed in EIS and DCP. However, the ionic conductivity value of cathode layer cannot be negligible ($\sim 0.1 \text{ mS cm}^{-1}$) because the ionic conductivity of SSE is 1.29 mS cm^{-1} through author's response. Could the authors clarify how the ionic conductivity is distinguished between SSE and cathode layer?

Response: We first apologize for our unclear explanation in the revised manuscript. Please allow us to clarify with additional data.

As shown in the EIS result, SSE-only and SSE-cathode configurations show comparable bulk resistances, and cathode-only configuration shows a negligible bulk resistance (Supplementary Fig. 19e). This low bulk resistance of cathode layer is due to the dominant contribution of electrical conductivity over ionic conductivity in electron non-blocking scenarios of cathode layer along with Li-blocking scenarios of Ti plungers. (Supplementary Fig. 19a, b, d). Similarly, DCP measurement also shows comparable results for both SSE-only and SSE-cathode configurations (Supplementary Fig. 19f). This suggests that the high electronic conductivity of cathode layer serves as an effective electron non-blocking layer, allowing to measure ionic and electronic

conductivities of SSE layer in an SSE-cathode configuration with comparable results to SSE-only configuration.

For ionic conductivity of cathode layer, the conventional electron blocking cell configuration was employed (Supplementary Fig. 19c). Specifically, for the SSE sandwich layers, powder SSE was used for freestanding films, and partial power SSE and SSE layer of co-rolled film were used for co-rolled film. As shown in the EIS result (Fig. 4e), the bulk resistance (R_1) is contributed to the ionic conductivity of sandwiched SSE layers, whereas the resistance of semi-circle ($R_2 + R_3$) is contributed to the ionic conductivity of the cathode layer, which was obtained by fitting the equivalent circuit in Supplementary Fig. 20b.

Supplementary Fig. 19 | Cell configurations and electrochemical methods used for the characterization of (a) Li⁺ transport in SSE, (b) e⁻ transport in SSE, (c) Li⁺ transport in cathode, and (d) e⁻ transport in cathode. Comparison of measurement results on (e) EIS and (f) DCP between SSE and SSE-Cathode configurations. Li⁺ and e⁻ transports in SSE were conducted with SSE-cathode structure due to the intrinsically integrated structure of co-rolled film. The high e⁻ conductivity of cathode layer ($> 30 \text{ mS cm}^{-1}$) showed negligible impacts on the bulk impedance of EIS for Li⁺ transport in SSE and the current measured of DCP for e⁻ transport in SSE. For Li⁺ transport in cathode, powder SSE was used for freestanding films, and partial power SSE and SSE layer of co-rolled film were used for co-rolled film. The overall thickness of SSE layers was fixed. For e⁻ transport in cathode, SSE layer of co-rolled film was carefully separated from cathode layer by peeling-off with tapes.

2. In Supplementary Fig. 23, the resistance from SSE-cathode interface is increased during charge. Authors defined the onset of resistance increase as “interface evolution”. How do authors confirm that the new interface between SSE and cathode is formed? Is there any probability that the contact loss between SSE and cathode due to the volume change of cathode particles? Alternatively, were the cells operated at sufficiently fast C-rate to rule out chemical degradation effects?

Response: We apologize again for the confusion in our revised manuscript. Please allow us to clarify and correct this.

The resistance contribution should be both the interphase (chemical degradation) and interface (contact loss). Because the oxidation potential of LPSCl (~ 2 V vs. Li/Li⁺) is lower than the operating potential of the cathode material, it is not possible to rule out the chemical degradation effect. However, the contact loss effect can be decoupled to some degree by varying the stack pressures. For example, the DRT analysis in Fig. 4i shows an increase of resistance from high stack pressure of 75 MPa to low stack pressure of 2 MPa. This resistance change with decreasing stack pressure is mostly contributed by the interface, where we found from cross-sectional SEM analysis (Fig. 5e) that it is highly relevant to the contact loss between SSE and cathode or interfacial voids.

In Supplementary Fig. 23, we initially wanted to highlight the interface change between co-rolled and freestanding films at low stack pressure. However, as the reviewer correctly pointed out, it is misleading to label the onset of resistance increase as “interface evolution”, as both interface and interphase evolve during first charging. Therefore, we have changed the label to “SSE-Cathode resistance” to avoid further confusion. We have modified the main text as below.

Supplementary Fig. 23 | In-situ observation of SSE-cathode resistance evolution of co-rolled films and freestanding films at 2 MPa in a half-cell configuration using LiIn as the counter

electrode. EIS measurement and DRT analysis, respectively, of (a, b) discharge process and (c, d) charge process of co-rolled film. EIS measurement and DRT analysis, respectively, of (e, f) discharge process and (g, h) charge process of freestanding films.

(Page 12, Main Text)

“The in-situ EIS-DRT analysis further confirmed the different behavior of their resistance evolution, where the cell assembled with co-rolled film maintained lower SSE-cathode resistance than that with freestanding films during charge/discharge at 2 MPa (Supplementary Fig. 23).”

3. In Supplementary Fig. 25, authors showed the pressurization test of LTO || NCM full-cell. While this experimental setup effectively demonstrates the optimized performance of the co-rolled film, I recommend including pressurization tests for freestanding films as well. This comparison could further emphasize the mechanical and operational advantages of the co-rolled films over freestanding counterparts. Given the expected more dramatic distance changes in freestanding films, such results would strongly support the importance of achieving a robust SSE-cathode interface through the co-rolling process.

Response: We appreciate the reviewer for the insightful suggestion!

As the reviewer recommended, we have provided pressure changes of co-rolled film and freestanding films (Supplementary Fig. 30). The cell with co-rolled film showed less pressure decrease than that with freestanding films. This result further supports the less formation of interfacial voids for co-rolled films and highlights the importance of constructing the robust interface. We have modified the main text as below.

Supplementary Fig. 30 | Monitored pressure change of (a) co-rolled film and (b) freestanding films with LTO anode at 75 MPa for 20 cycles. The cell with co-rolled film showed less pressure decrease than that with freestanding films (approximately -2 MPa vs. -3 MPa).

(Page 14, Main Text)

“This result explains that the capacity loss observed with lower stack pressure for freestanding films can be attributed to more void formation at SSE-cathode interface which increases the polarization and lowers capacity utilization⁵⁵. The less pressure change of the cell with co-rolled film during long-term cycling further supports the less formation of interfacial voids compared to that with freestanding films (Supplementary Fig. 30).”

Reviewer #3

The authors have revised the manuscript very carefully. All comments have been well addressed. It can be accepted by Nature Commun in the current form.

Response: We appreciate the reviewer's positive feedback on our work and acceptance of publication.